# Active-site loop variations adjust activity and selectivity of the cumene dioxygenase

Peter M. Heinemann [1], Daniel Armbruster [1] & Bernhard Hauer [1✉]

Active-site loops play essential roles in various catalytically important enzyme properties like activity, selectivity, and substrate scope. However, their high flexibility and diversity makes them challenging to incorporate into rational enzyme engineering strategies. Here, we report the engineering of hot-spots in loops of the cumene dioxygenase from *Pseudomonas fluorescens* IP01 with high impact on activity, regio- and enantioselectivity. Libraries based on alanine scan, sequence alignments, and deletions along with a novel insertion approach result in up to 16-fold increases in activity and the formation of novel products and enantiomers. CAVER analysis suggests possible increases in the active pocket volume and formation of new active-site tunnels, suggesting additional degrees of freedom of the substrate in the pocket. The combination of identified hot-spots with the Linker In Loop Insertion approach proves to be a valuable addition to future loop engineering approaches for enhanced biocatalysts.

[1] Institute of Biochemistry and Technical Biochemistry, Department of Technical Biochemistry, University of Stuttgart, Stuttgart, Germany. ✉email: bernhard.hauer@itb.uni-stuttgart.de

Members of one enzyme family often share the same basic scaffold with similar key catalytic residues, but a large diversity in selectivities, activities and substrate scope[1]. Nature achieves this diversity by minor changes, e.g., point mutations in the active-site, but also by major rearrangements in the protein structure and dynamics[2]. This last-mentioned evolutionary strategy mainly affects the alteration of exterior loops, usually the most flexible parts of proteins. These often low conserved regions contribute to various dynamic processes like substrate binding, directing processes into the active pocket, or conformational changes[3]. The rational engineering of loops gained attention in recent years generating improved or even novel biocatalysts[3,4]. Park et al. introduced β-lactamase activity into the metallohydrolase scaffold of glyoxalase II by remodeling several active-site loops. Based on corresponding loops in metallo-β-lactamases, the resulting variant gained hydrolysis activity for the previously not accepted substrate cefotaxime[5]. Loops located at the surface are often addressed for stability reasons[6–8]. Feng et al. demonstrated an 11.8 times longer half-life at 70 °C of variants of the β-glucuronidase from *Penicillium purpurogenum* Li-3 by loop transplants based on two thermophilic glycosidases[9]. Changes in selectivities can also be achieved by loop modulations. Based on structural analysis and molecular-dynamics (MD) simulations, the groups of Nie and Sun increased the volume of the binding pocket of a ketoreductase[10]. The mutation of two key residues located on a loop resulted in increased enantioselectivities in the reduction of bulky ketones to the corresponding pharmaceutically-interesting alcohols. Another interesting finding is the regioselectivity switch of nitrating P450s by a single point mutation in the F/G loop by the group of Arnold[11]. These examples, among many others, reveal the significant importance of loops for modulating enzyme properties[3].

Rieske non-heme iron-dependent oxygenases (ROs) are a class of enzymes primary known for the incorporation of molecular oxygen in aromatic compounds, resulting in *cis*-dihydroxylated, non-aromatic products[12,13]. ROs are multicomponent systems consisting of a NAD(P)H-oxidizing reductase, an oxygenase and often a ferredoxin for electron transfer[14]. The oxygenase comprises catalytically-active α-subunits and usually a corresponding number of structurally important β-subunits that form a $(\alpha_x\beta_x)$-heteromer. The α-subunit contains the Rieske-type [2Fe-2S] iron-sulfur cluster as an electron acceptor and an active-site non-heme iron[14]. Their large substrate scope, containing aromatics[15], heterocycles[16], olefins[17], and terpenes[17,18], combined with their broad reaction spectrum of *cis*-dihydroxylations[15], allylic monohydroxylations[17], desaturations[19], and sulfoxidations[20] makes ROs attractive biocatalysts for the synthesis of biologically-active products and fine chemicals[21–23]. Although known for a long time for their interesting features, ROs were not subject of such intensive engineering approaches as the related P450s[24]. The group of Gibson was among the first to evaluate the influences of specific active-site residues of the naphthalene dioxygenase (NDO) from *Pseudomonas* sp. strain NCIB 9816-4 and the toluene dioxygenase (TDO) from *Pseudomonas putida* F1 on activity and selectivity[25–27]. Since then, in various studies selectivities and activities were altered along with the broadening of the substrate scope, mainly focusing on the active site[28–31]. Ferraro et al. compared structurally analogous residues in the active-sites of eight oxygenases with published crystal structures[13]. They showed a high conservation of the same or similar amino acids, with 71.5% of them being hydrophobic, not taking the α-subunit-bridging or iron-coordinating residues into account. While the structural comparison of active sites and whole α-subunits from different ROs showed high similarities, active-site decorating loops differ significantly even between closely related oxygenases[13,14,32–35]. These flexible loops, located at the entrance of the active pocket, were often referred to conformational changes

during substrate binding and thereby directly influencing the substrate specificity[13,32,36,37]. This interesting feature was explored several times in the well-described biphenyl dioxygenases, but was also observed for other ROs[36,38–40]. Kim et al. investigated the *N*-demethylating ROs NdmA and NdmB from *Pseudomonas putida* CBB5, which utilizes caffeine as carbon source with the RO-catalyzed demethylation as the first step of degradation[39]. The crystal structure of NdmA showed a high similarity to several other ROs, including the biphenyl dioxygenase, the 3-Ketosteroid 9α-Hydroxylase (KSH) and the dicamba monooxygenase (DMO). One loop of the NdmA corresponds to a region in the DMO responsible for conformational changes[34,39,41]. Comparable to this, the loop in NdmA tightly closed the active site by hydrophobic interactions in the presence of substrate. A NdmA variant, derived from loop grafting with NdmB, transferred the NdmB activity. Capyk et al. characterized the KSH from *Mycobacterium tuberculosis* which plays a central role in its steroid catabolism[33]. They identified the opening of the active site tunnel between the helix α5 and a loop between the strands β15 and β16[33]. Petrusma et al. identified the same loop as responsible for the substrate preference and altered this by the construction of chimeric loop variants based on KSH homologs[40]. In addition, mutations in these chimeras increased the activity for multiple substrates. Another striking example that underlines the influence of active-site loops on the properties of ROs as biocatalysts is the occurrence of 19 different oxygenases in the genome of *Phenylobacterium immobile* E[42]. This organism utilizes man-made compounds like pyrazone or phenazone as sole carbon source, with oxygenases catalyzing the crucial dihydroxylation of the aromatic moiety as the first step of degradation[43]. Remarkably, they differ mainly by only one sequence region. The sequence alignment of these α-subunits with literature-known dioxygenases reveals that this low conserved sequence space corresponds to loop 1 in the CDO and a related loop-coding sequence in the NDO (Supplementary Fig. S1)[44]. Furthermore, all aligned oxygenases share a low conservation for the sequence space coding for loop 2 in the CDO, while both putative loop sequences are framed by highly conserved motifs (Figs. S1 and S7). Proteome analysis of the organism grown on different carbon sources revealed differing expression levels of the 19 oxygenases, depending on which substrate they were cultivated on[45].

Herein, we want to employ different loop engineering strategies for the diversification of a biocatalyst to alter important aspects like substrate scope, activity and selectivities. Two loops above the active-site (loop 1 and 2, Fig. 1) of the cumene dioxygenase (CDO) from *Pseudomonas fluorescens* IP01 were subjected to the identification of novel hot-spot positions and the evaluation of different insertion and deletion (InDel) libraries. Well-chosen substrates represent different compound classes, as well as offering further insights in changes in chemoselectivity and regioselectivity. In the course of the different applied engineering strategies, loop 2 was identified as a hot-spot region with numerous improved variants. The evaluation of the different loop engineering approaches, including the novel Linker In Loop Insertion (LILI) approach based on protein linkers, yielded various variants with significant changes in product formation and distribution. Finally, a CAVER analysis suggested the possible broadening of already existing and the formation of new tunnels.

## Results

**Identification and saturation of loop hot-spots.** Compared to the common planar substrates of other ROs, cumene as the natural substrate of the CDO from *Pseudomonas fluorescens* IP01 is sterically more demanding, suggesting a larger active-site pocket and thus, an ideal starting point for further engineering approaches[14].

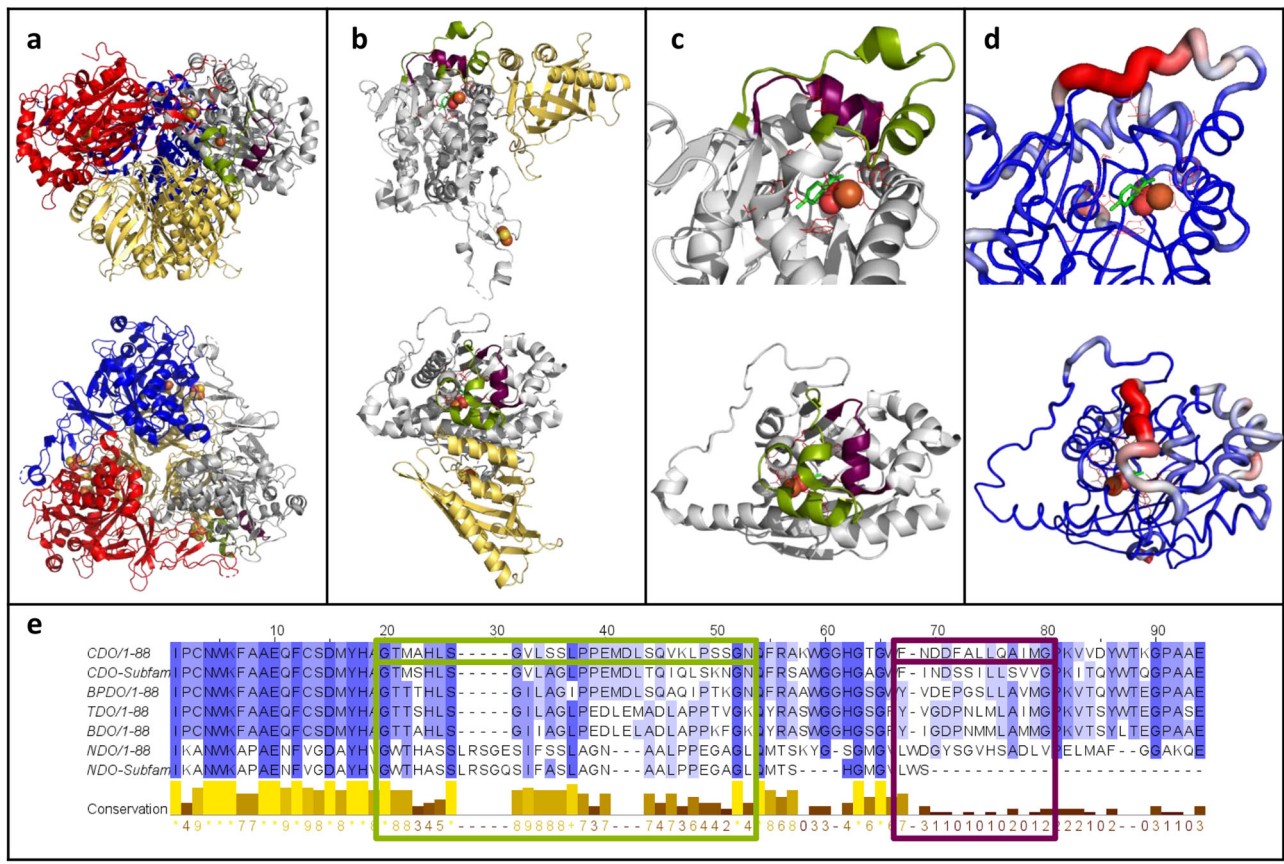

**Fig. 1 The selected loops 1 and 2 of the cumene dioxygenase from *Pseudomonas fluorescens* IP01 (PDB entry: 1WQL). a** Side view (top structure) and top view (bottom structure) of the hexameric quaternary structure oxygenase consisting of three α-subunits (gray, blue and red) and three β-subunit (yellow). Loop 1 (29 residues, green), loop 2 (13 residues, purple), as well as the catalytic iron (orange sphere) and the Rieske center (yellow and orange spheres) are shown. **b** Side view and top view of the dimeric structure of the α-subunits (gray) and the corresponding β-subunit (yellow) with loop 1, loop 2, catalytic iron and Rieske cluster in the same color code like **1a**. **c** Side view and top view of an enlargement of loop 1 (green) and loop 2 (purple) in the crystal structure of one α-subunit (gray) of the oxygenase of the CDO. The catalytically active iron (orange sphere), molecular oxygen (red spheres) and active-site residues (red sticks) are shown, as well as the docked substrate (*R*)-limonene (green sticks). **d** B-factor indicated by color-coded cartoon of the crystal structure from low (blue) to high (red) from the same angles like **c**. Iron, molecular oxygen, (*R*)-limonene and active-site residues are shown in the same color pattern as in **1a**. **e** Sequence alignment of the α-subunit of the oxygenases from CDO, the CDO subfamily, biphenyl dioxygenase (BPDO) from *Paraburkholderia xenovorans* (LB400), toluene dioxygenase (TDO) from *Pseudomonas putida* F1, benzene dioxygenase (BDO) from *Pseudomonas putida* ML2, naphthalene dioxygenase (NDO) from *Pseudomonas* sp. NCIB 9816-4 and the NDO subfamily. The degree of conservation is shown from low (white) to high (blue), as well as in the conservation line. Loop 1 (green), loop 2 (purple), and the corresponding residues in the other ROs are highlighted by frames in the corresponding colors.

Previous studies showed allylic monohydroxylation, as well as dihydroxylation of a broad substrate scope including aromatics and monoterpenes, catalyzed by the CDO[17]. Based on literature reports and structures, as well as sequence comparisons of the α-subunit of the CDO oxygenase with other ROs, we selected the two loops for this study, both located at the outer surface of the quaternary structure (Fig. 1 and Supplementary Fig. 1)[14]. The larger loop 1 (residues G236-N264, Fig. 1c/d) contributes mainly to an active-site tunnel while implying to form a cap over the buried active-site (Supplementary Fig. 3)[14,46,47]. The b-factor values of loop 2 (F278-G290, Fig. 1c/d) with up to 31.7 Å² for the backbone of L284 indicates a high flexibility of the α-helix, compared to an average b-factor of 24.6 Å² for the loop 2 backbone, or the average b-factor of 21 Å² for the whole α-subunit[14]. The b-factor values of the backbone of loop 1 peak in P260 with 57.6 Å² (whole loop: 36.9 Å²), the highest value of the whole α-subunit[14]. A sequence alignment with oxygenases from previously described ROs, like TDO, NDO, and novel ROs from *Phenylobacterium immobile* E, indicates very low conservation of the loops that are framed by highly conserved motifs (Fig. 1e and Supplementary Fig. 1). The 42 loop residues

were subjected to an alanine scan, with wild-type alanine residues mutated to glycine. The generated variants were tested with styrene **1**, (*R*)-(+)-limonene **2**, and 2-phenylpyridine **3** as substrate panel, illuminating various aspects of the new biocatalysts. They represent the different compound classes small aromatics (styrene), monoterpenes ((*R*)-(+)-limonene), and *N*-heterocycles (2-phenylpyridine). Furthermore, they were supposed to give further insights into changes of the chemoselectivity (allylic monohydroxylation vs. dihydroxylation), regioselectivity (aromatic ring vs. alkene side chain) and activity increases for a poorly accepted substrate (2-phenylpyridine). Results are shown in Fig. 2, Supplementary Figs. 4–6 and Supplementary Table 1.

From 42 tested variants only three, including the iron-coordinating H240A, were inactive. In contrast, various variants showed strong influences on product formation and distribution. Especially the variants of loop 2 revealed drastic changes (Fig. 2), despite up to 20 Å from D280 to the iron. Single-point variants F278A, N279A, and D280A enhanced product formations up to 32.4% for D280A with substrate **1**. Compared to the wild-type, differing product selectivities ranging from 93.6% **1a** for F282A to

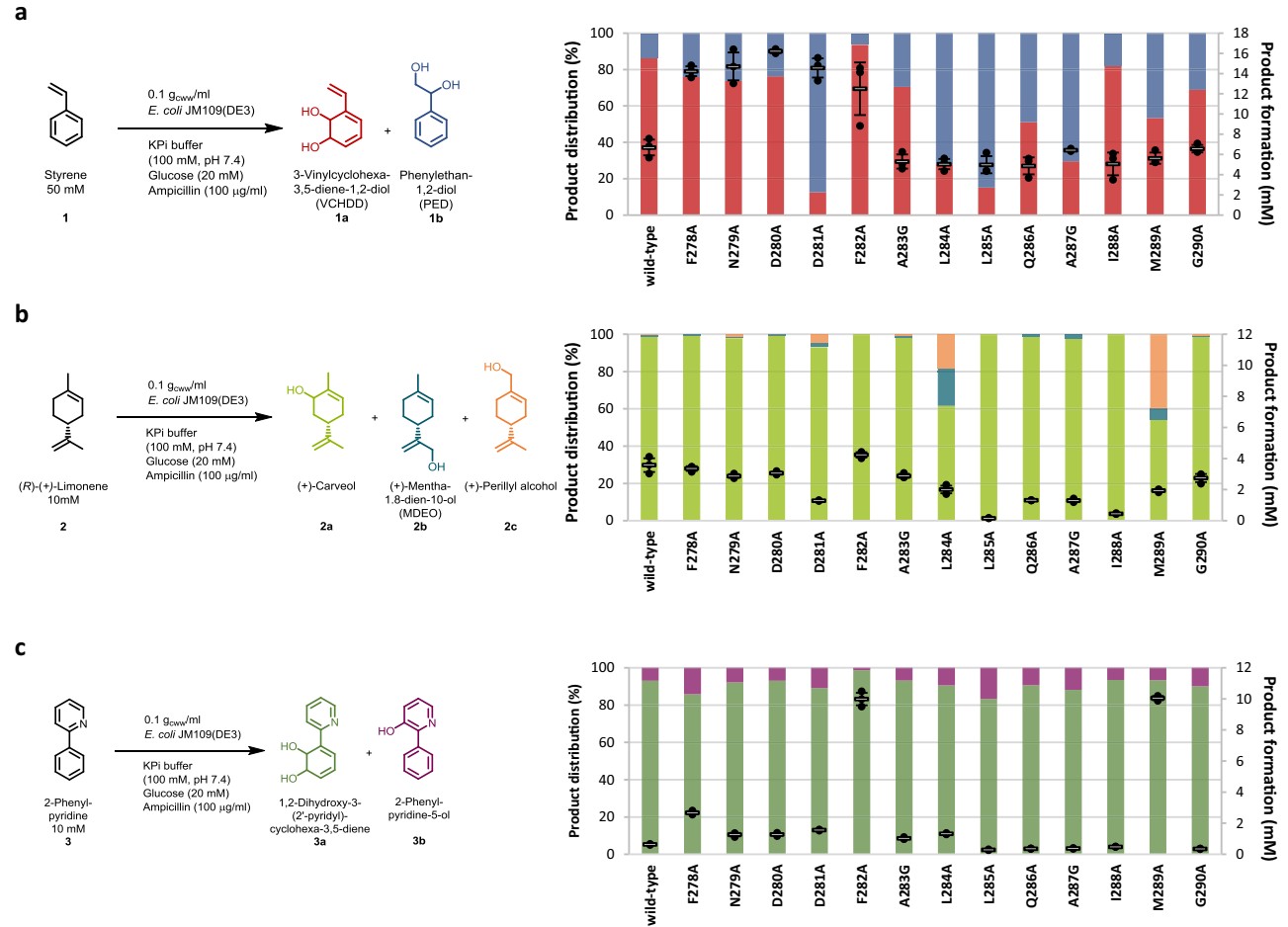

**Fig. 2 Biotransformations with the wild-type and alanine variants of loop 2. a** Product formation and distribution of the products 3-vinylcyclohexa-3,5-diene-1,2-diol **1a** (VCHDD, red, bottom bars) and phenylethan-1,2-diol **1b** (PED, blue, top bars) obtained during the biotransformation of styrene **1**. **b** Product formation and distribution of the products (+)-carveol **2a** (green, bottom bars), (+)-mentha-1.8-dien-10-ol **2b** (MDEO, blue, middle bars) and (+)-perillyl alcohol **2c** (orange, top bars) obtained during the biotransformation of (R)-(+)-limonene **2**. **c** Product formation and distribution of the products 1,2-dihydroxy-3-(2′-pyridyl) cyclohexa-3,5-diene **3a** (green, bottom bars) and 2-phenylpyridine-5-ol **3b** (purple, top bars) obtained during the biotransformation of 2-phenylpyridine **3**. Reaction conditions and substrate concentrations are indicated in the reaction equation. The reactions were performed in technical triplicates (black dots) with average values (horizontal bar) and standard deviations (calculated using Excel version 2016) indicated. Error bars may be covered by markers. Source data are provided as a Source Data file.

87% **1b** for D281A were obtained (Fig. 2a). The variants F282A and M289A showed the most drastic changes regarding product formation and regioselectivity for **2**. F282A increased as the only variant the product formation, with **2a** as the single product, while M289A enhanced noticeably the amount of the product **2c** up to 40%. The biotransformation of **3** underlines the influence of loop 2 in general and the residues F282 and M289 in particular regarding an increased product formation like shown before for **2** (Fig. 2c). Both variants led to improved product formations up to almost 17-fold for F282A and M289A compared to that of the wild-type.

Based on these results, the residues F278 and D280 to M289 were subjected to single site saturation mutagenesis via NNK-codon and subsequent testing of **2** as screening substrate in 96 deep-well plates. 92 colonies per site saturated position were picked and tested to achieve a theoretical library coverage of almost 95%. The eight most interesting variants were re-evaluated in 1 ml scale in glass vials with 50 mM of substrate **1** and 10 mM of the substrates **2**-**3** (Supplementary Table 2). The low product formation of all variants for **1** and except for one variant also for **3** was not surprising due to their adaption to **2**. F282T showed a remarkable increase in product formation to 98.2% for **3**, with

improved regioselectivity up to 97% **3a**. For **2**, F278V and F282V increased the product formation to 53% and 52%, respectively, with regioselectivity of over 99% **2a**. As observed previously in the alanine scan and additionally by the variant L284G derived from the site saturation mutagenesis, residue L284 seems to influence the substrate positioning due to the highest formation of **2b** and **2c** for all tested variants. L284G showed a decreased product formation of only 21% with a shift in selectivity to almost 50% **2c**.

The tested variants with single-site mutations demonstrated the impact of loop 1 and loop 2 on activity and selectivity. This classical approach gave hints on new hot-spot residues like F278, F282, and M289. Nevertheless, it was not clear if the introduced mutations interact directly with the substrate or alter the loop dynamics, so we wanted to evaluate further approaches of loop engineering, focusing on the loop length and thereby its dynamics. Various examples from the literature suggest that the modulation of loop flexibilities could alter specific enzyme properties such as substrate specificity, stability or activity[3,4]. Therefore, additional engineering approaches like InDels could lead to equivalent or even better biocatalysts compared to the alanine scan and saturation variants.

**Evaluation of an adaption library based on new oxygenases**. A brief look into Nature's evolution strategies highlights that major evolutionary steps are often achieved by InDels[1,48]. This strategy was tested for CDO by comparing the loop sequences with corresponding sequences of ROs found in the genome of evolutionarily distant *Phenylobacterium immobile* E (Supplementary Fig. 2)[42]. This approach is similar to the stepwise loop insertion strategy (StLois) of Hoque et al.[49]. They inserted residue pairs with degenerate NNK codon randomization in a stepwise manner at two loop positions in the Phosphotriesterase-like lactonase from *Geobacillus kaustophilus* (*Gka*P-PLL). This loop corresponds to loop 7 in the related hydrolyzing Phosphotriesterase *pd*PTE from *Pseudomonas diminuta*, playing a crucial role in substrate specificity[50]. These two enzymes share a similar folded structure. However, the loop in *Gka*P-PLL is eleven residues shorter than loop 7 of the *pd*PTE. Loop insertions in *Gka*P-PLL resulted in drastic increases in activity and substrate specificity for the hydrolysis of organophosphate substrates, undergoing a shift from lactonase to hydrolysis activity. The comparison of loop 1 and 2 of the CDO with the corresponding loops of the oxygenases of *Phenylobacterium immobile* E clearly shows that the CDO-loops are significantly shorter, but framed by highly conserved motifs, comparable to the loops from the study of Hoque[49]. In order to have a lower screening effort and to gain insight how the insertion composition influences the enzyme properties, sequence alignment-based insertion pairs were introduced (Supplementary Fig. 2). Two libraries were created: The first one introduces single-point mutations and deletions of highly conserved residues into the loops of the CDO, based on the consensus sequence (19 variants, Supplementary Fig. 7, red and green). The second library is based on stepwise insertions of the residues of the in each case two longest loops that correspond to loop 1 and 2 (26 variants, Supplementary Fig. 7, yellow). These residues were inserted stepwise to "fill up" the loops of the CDO. The nomenclature of the variants follows the widely accepted suggestion of den Dunnen and Antonarakis[51]. The naming of insertion variants is based on the position of the two framing residues, followed by "ins" and the insertion. The nomenclature of deletion variants follows either the rule of naming the deleted residue when a single deletion was applied, or mentioning the first and last deleted residues, followed by "del". The libraries were again evaluated with the substrates 1–3 (Supplementary Tables 3 and 4). The majority of the single-point mutations and deletions based on the *Phenylobacterium immobile* E oxygenases showed no improvements compared to the wild-type. The only exceptions were the variants F278G and D281E with increased product formations for 1 and 3 (Fig. 3). This result is consistent with the observations from the alanine scan, when mutations of these positions resulted in altered selectivities and product formations. Remarkably, only 19% of the insertion variants were inactive, despite up to nine inserted amino acids. Loop 2 confirmed to be a hot-spot region. Multiple variants, with insertions ranging from one to seven amino acids, showed clear improvements in activity and selectivity (Fig. 3).

**Deletion of loop residues improves the CDO biocatalyst**. The sequence alignment-based deletions resulted in variants with strongly decreased product formations. It should be noted that these deleted positions were located in a conserved region in the direct periphery of the catalytically iron. In contrast, deletions in flexible and barely conserved loops could remodel them and have a positive effect on enzyme properties[5]. Therefore, further deletion variants should be tested to evaluate if deletions could be a suitable part of loop engineering. We tested nine different variants with one to seven lacking amino acids. The target residues

were rationally picked by selecting the approximate center of loop 1 (E251), the previously described interesting residue V257, and the center of loop 2 (A283). These residues were deleted separately to construct the first three deletion variants. The next three deletion variants were constructed by deleting the two neighboring amino acids of the previously removed residues, resulting in three variants with three deleted amino acids. The last three deletion variants were constructed by deleting the two neighboring amino acids of the previously deleted residue, thereby resulting in deletion variants with seven removed residues. The removal of seven amino acids as a large section of loop 1 or almost the complete loop 2 resulted in the loss of activity (Fig. 3 and Supplementary Table 5). Deletions in loop 1 increased product formations and selectivities slightly for 1b and decreased the formation of the products with stable product distributions for the other substrates. Surprisingly, the deletion of one or three amino acids in the middle of loop 2 yielded two of the variants with the highest product formations for 2 and 3. The single residue deletion of A283del increased the activity, especially 89.2% product formation for 2 with high product selectivity of 98.2% 2a (0.8% 2b, 1.1% 2c) and >99% product formation for 3 with 98.7% 3a. The variant F282_L284del showed a product formation of 39.9% for substrate 1, the second highest of all tested variants, with a regioselectivity of 93.7% 1b, what represents a switched product distribution compared to the wild-type.

These previously tested biocatalysts emphasize deletions and insertions as useful tools for loop engineering, in particular at the beginning and at the end of loop 2. A considerable disadvantage of this technique is the lack of insights into the impact of the mutated amino acids; if changes in the loop flexibility and thereby changes in the shape of the active pocket, or individual interactions of the side chains of introduced amino acids, like formed H-bonds or ionic interactions, contribute to altered catalytically important enzyme properties. The flexibility of loops makes it difficult to predict beneficial mutations, especially without simulations. Stepwise insertions of amino acids with known dynamic properties can yield a deeper understanding how altered loop agilities could affect enzyme properties. Therefore, we applied the Linker In Loop Insertion (LILI) approach, based on stepwise insertions of protein linkers with known properties at hot-spot positions[52]. Although this approach is similar to the approach of Hoque et al., the screening effort is considerably lower and significant changes in the loop dynamics can be achieved with only a few insertions[49]. Thus, conclusions can be drawn about beneficial or counterproductive changes in enzyme properties based on modifications of loop length and flexibility.

**Insertion of protein linkers as loop engineering strategy**. The insertions, as well as the single-point mutations based on sequence comparison to *Phenylobacterium immobile* E oxygenases revealed the positions V257, F278, and Q286 with influence on enzyme properties and high tolerance towards insertions and were therefore targeted for the evaluation of our approach. Four examples of linkers with known flexibility properties were selected: the flexible (G) and (GS), the stiff and non-helical (PA) and the polar (GP) linker without a certain secondary structure (Fig. 4)[52–56]. Two to six linker residues were inserted after V257, F278, and Q286, and the resulting libraries were tested with the substrates 1–3 (Supplementary Fig. 11, Tables 6, 7, and 8).

In comparison to the alignment-based library, LILI variants revealed interesting trends based on the length and composition of the insertions (Fig. 5a). Longer insertions after F278 tend to increase the formation of 1b, like the mutation F278_N279insGS resulted in a compared to the wild-type slightly increased product formation of 28.5% simultaneously with an increase of 1b to

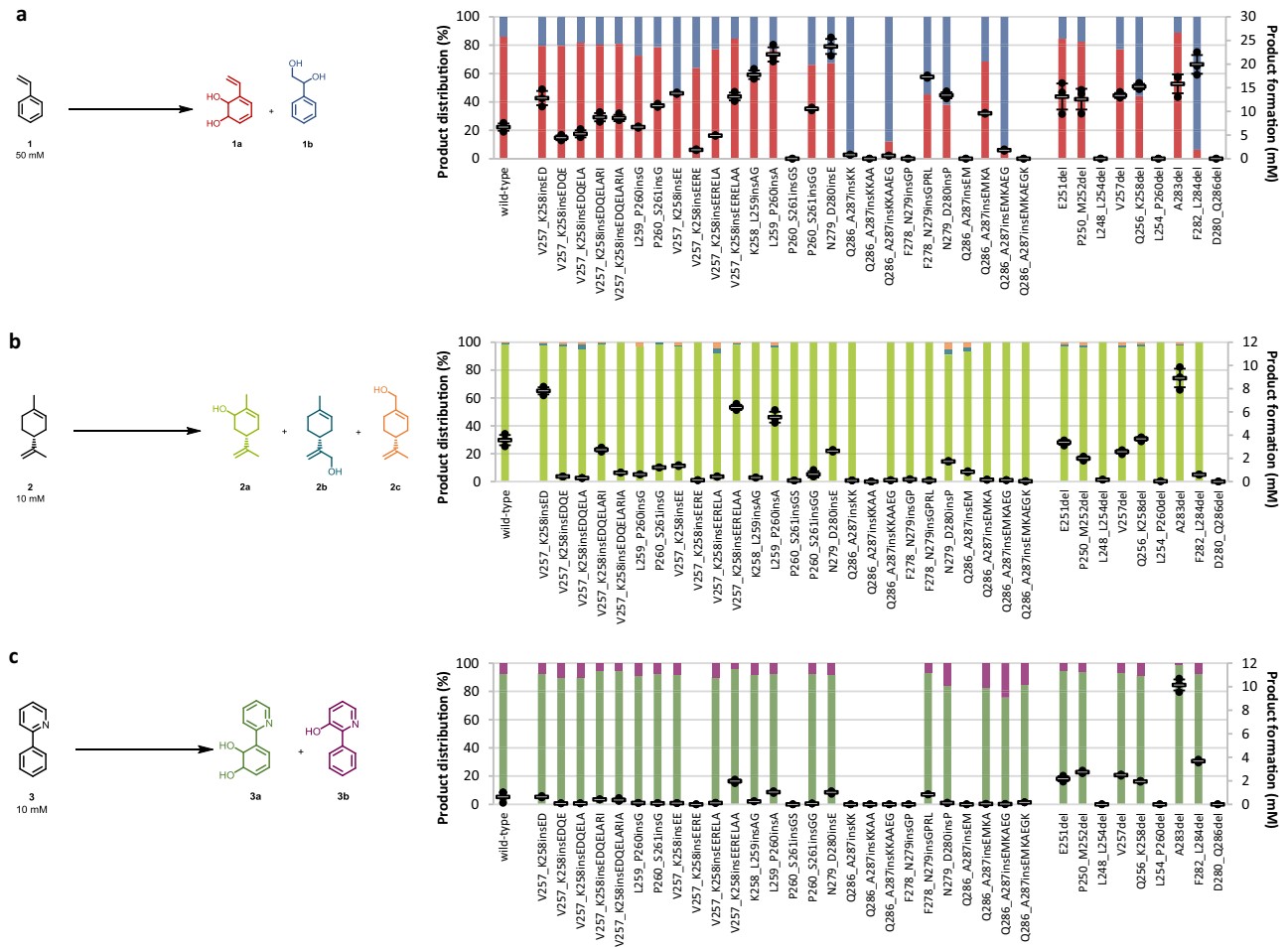

**Fig. 3 Biotransformations with the wild-type, insertion and deletion variants.** Insertion variants were derived from the sequence alignment with oxygenases from *Phenylobacterium immobile* E. **a** Product formation and distribution of the products 3-vinylcyclohexa-3,5-diene-1,2-diol **1a** (VCHDD, red, bottom bars) and phenylethan-1,2-diol **1b** (PED, blue, top bars) obtained during the biotransformation of styrene **1**. **b** Product formation and distribution of the products (+)-carveol **2a** (green, bottom bars), (+)-mentha-1.8-dien-10-ol **2b** (MDEO, blue, middle bars) and (+)-perillyl alcohol **2c** (orange, top bars) obtained during the biotransformation of (R)-(+)-limonene **2**. **c** Product formation and distribution of the products 1,2-dihydroxy-3-(2'-pyridyl) cyclohexa-3,5-diene **3a** (green, bottom bars) and 2-phenylpyridine-5-ol **3b** (purple, top bars) obtained during the biotransformation of 2-phenylpyridine **3**. Reaction conditions and substrate concentrations are indicated in the reaction equation. The reactions were performed in technical triplicates (black dots) with average values (horizontal bar) and standard deviations (calculated using Excel version 2016) indicated. Error bars may be covered by markers. Source data are provided as a Source Data file.

57.5%. The activity rises corresponding to the insertion length up to 40.6% product formation with F278_N279insGSGSG with 66.5% **1b**. The two longest insertions of the respective sets, "PAPAPA" and "GPGPGP", also yielded the highest proportion of **1b**, although with a lower product formation. Interestingly, LILI-based insertions after Q286 resulted in variants with **1b** as the predominant or the sole product, regardless of the length of the insertion. Comparable activities to insertions after F278, with even further increased regioselectivity for **1b**, could be observed with Q286_A287insGSGSG with 34.1% product formation and >99% **1b** or Q286_A287insPAPAPA with 37.9% product formation and 92.9% **1b**. Interestingly, a single inserted amino acid can make the difference between increased activity and none. The insertion of "GS", "GSG", and "GSGSG" after Q286 increased the product formation for **1**, while "GSGS" and "GSGSGS" resulted in a loss of activity for all of the tested substrates (Supplementary Fig. 11). The evaluation of LILI variants after F278 with **2** resulted in various variants with slightly enhanced activity, as well as increased amounts of **2b** and **2c** (Fig. 5b). However, the effects were to a clearly smaller extent than for substrate **1**. The activity peaks with variant F278_N279insGP

(69.2% product formation), while longer GP insertions caused lower product formations but reduced the regioselectivity up to 6.8% **2b** with F278_N279insGPGP or 8.8% **2c** with F278_N279insGPGPG, compared to the 0.7% **2b** and 0.5% **2c** for the wild-type. Insertions after Q286 resulted in lowered product formations with only slightly increased amounts of **2b** and **2c**. The LILI insertions, especially the uncommon "PA" and "GP" sets after F278, showed a high impact on the transformation of the bulky substrate **3** (Fig. 5c). The product formation was enhanced from 6.4% with the wild-type up to 61.6% with F278_N279insPAPA (96% **3a**) and even 74.2% with F278_N279insGP (96% **3a**). Consistent with the results from the biotransformations with sequence alignment-based insertions based on *Phenylobacterium immobile* E, the LILI-based insertions after Q286 influence the regioselectivity, since all variants showed an increase of **3b**. The amount of **3b** peaked in Q286_A287insGPG with up to 20.2% **3b** and 18.6% product formation. Interestingly, most of the LILI insertions after V257 strongly decreased the activities compared to the previously applied insertions at this position. While LILI-based insertions after F278 resulted in more variants with enhanced properties, the

Group 1:
GG
GGG
flexible
GGGG
GGGGG
GGGGGG

Group 2:
GS
GSG
flexible
GSGS
GSGSG
GSGSGS

Group 3:
PA
PAP
stiff, non-helical
PAPA
PAPAP
PAPAPA

Group 4:
GP
GPG
no defined
GPGP
structure, polar
GPGPG
GPGPGP

**Fig. 4 The insertion sets of the LILI library.** The four sets of applied linkers as insertions after V257, F278, and Q286 for the LILI library. Group 1 and 2 represent very flexible, polar linkers. Group 3 is known as stiff and non-helical linkers. Group 4 insertions are polar and show no distinct structure.

Q286_A287ins-variants show the applicability of the approach at a second insertion site, resulting in improved enzymes, especially for the formation of **1b**.

**Further insights into the loop variants**. The biocatalysts with the highest activities and selectivities for the individual products from this study are listed in Supplementary Table 9. Various selected variants were subjected to the determination of the *ee* values (Supplementary Table 10). No change of the already high *ee* of >99% (+)-(1*S*,2*R*)-**3a** of the wild-type with **3** was observed. Most of the variants showed excellent *ee* values of >99% (1*R*,5*S*)-**2a**, comparable to 99% (1*R*,5*S*)-**2a** of the wild-type. A slight decrease of the *ee* to 93% (1*R*,5*S*)-**2a** was monitored with variant F282_L284del. More drastic changes were noticed with **1**. The wild-type showed a low selectivity of 18.4% (*R*)-**1b**. F282A increased the selectivity for (*R*)-**1b** up to 72.7% (*R*)-**1b**, while I288S inverted the enantioselectivity to 15.6% (*S*)-**1b**. Despite using (*R*)-**2** at a purity of 97%, impurities of (*S*)-**2** are inevitable. Therefore, peaks of (1*R*,5*R*)-**2a** were observed which enables the determination of the diastereomeric excess (Supplementary Table 11). While the wild-type shows a high *de* of 94.7% (1*R*,5*S*)-**2a**, the variants F282_L284del and L284G decreased the *de* to 55.1% and even 38.1% (1*R*,5*S*)-**2a**, respectively.

The computational tools YASARA and CAVER were used to gain insights how the mutations could have altered active-site tunnels. These tunnels play crucial roles in access and egress processes to the active-site and the positioning of substrates[3,47,57]. This critical influence has already been demonstrated for the naphthalene dioxygenase by Escalante et al. or for two Rieske monooxygenases involved in the biosynthesis of paralytic shellfish toxins by Lukowski et al.[37,58]. Loops contribute significantly to the shape of these tunnels, so it is worth investigating if the loop

mutations alter them. Homology models of the eight loop variants listed in Supplementary Table 9 were created. The wild-type and the loop variants were subjected to 500 ps MD refinements and subsequent short MD simulations of 5 ns[57,59–61]. The received average structures of the simulation steps were then analyzed via CAVER (Fig. 5)[47]. Due to the fact, that these simulations are only single replica and the simulation time of 5 ns is relatively short, the resulting structures cannot illustrate the exact structural outcome of the mutations on the shape of the loop and tunnels[61,62]. Nevertheless, the resulting models and CAVER analyses could be a hint about the influence of the mutations. The short simulations showed two tunnels in the wild-type, a broad one, previously mentioned by Dong et al. and highlighted in green in Fig. 5, as well as a more narrow tunnel, never described before, highlighted in red[14]. Even small modifications like the point mutations in F282A, M289A, and L284G, suggest changes in the shape and volume of the cavity and even the formation of a new tunnel like in M289A. These altered pocket shapes could provide additional degrees of freedom during substrate positioning. This assumption is backed up by the tremendous changes in the regioselectivity, especially for the biotransformation of **2** with M289A and L284G. While the influence of the point mutations is mainly observable in the formed tunnels, insertions and deletions also change noticeably the shape of the loops, even though the applied simulations cover only a short period of time and thereby rather small loop movements. The insertions after Q286 and the deletion of three amino acids in F282_L284del decreased the length of the α-helix in loop 2 and changed the positioning the lid-like loop 1. The LILI-library based variant Q286_A287insGSGSG opened an additional tunnel, suggesting a possible new entrance route for substrates, as well as cosubstrates. F282A and A283del, both variants with strongly increased product formations compared to the wild-type, seem to not alter the direction of the main tunnel tremendously, but widened the pocket and thereby facilitate substrate access and product egress. The results of the CAVER analysis suggest how loop modulations could alter active site tunnels and provide first hints how the incorporated mutations influence catalytically important properties of the enzymes despite their big distance to the iron. Future simulations with a significantly larger number of replicas and longer simulation durations will provide deeper and more accurate insights into changes in the dynamics and shape of the mutated loops and tunnels that go far beyond the suggestive character of these simulations.

## Discussion

Inspired by Nature and, due to continuously improving MD simulations, a growing understanding of how altered loop flexibilities and compositions influence different aspects of biocatalysts, the engineering of active-site loops is becoming an increasingly used tool. It is still challenging to predict precisely beneficial loop modulations due to their high flexibility. This complexity makes it inevitable to apply well-known and new engineering approaches to these relatively unexplored structure elements. In this study, we examined alanine scan, sequence alignment-based mutations, and novel InDels on two loops of the CDO. The conservative strategy of exchanging the residues with alanine, respectively, alanine in the wild-type by glycine, revealed various hot-spots not yet described (loop 2), probably due to the long distance to the non-heme iron. Mutations of F282 and M289, highlighted positions in Fig. 6, showed drastic influences on mainly the product formation, especially for the more bulky substrate. I288 and L284 seem to influence the positioning of the substrate and, thereby altering the regioselectivity and enantioselectivity (Fig. 7). InDels, influencing flexibility and length of the

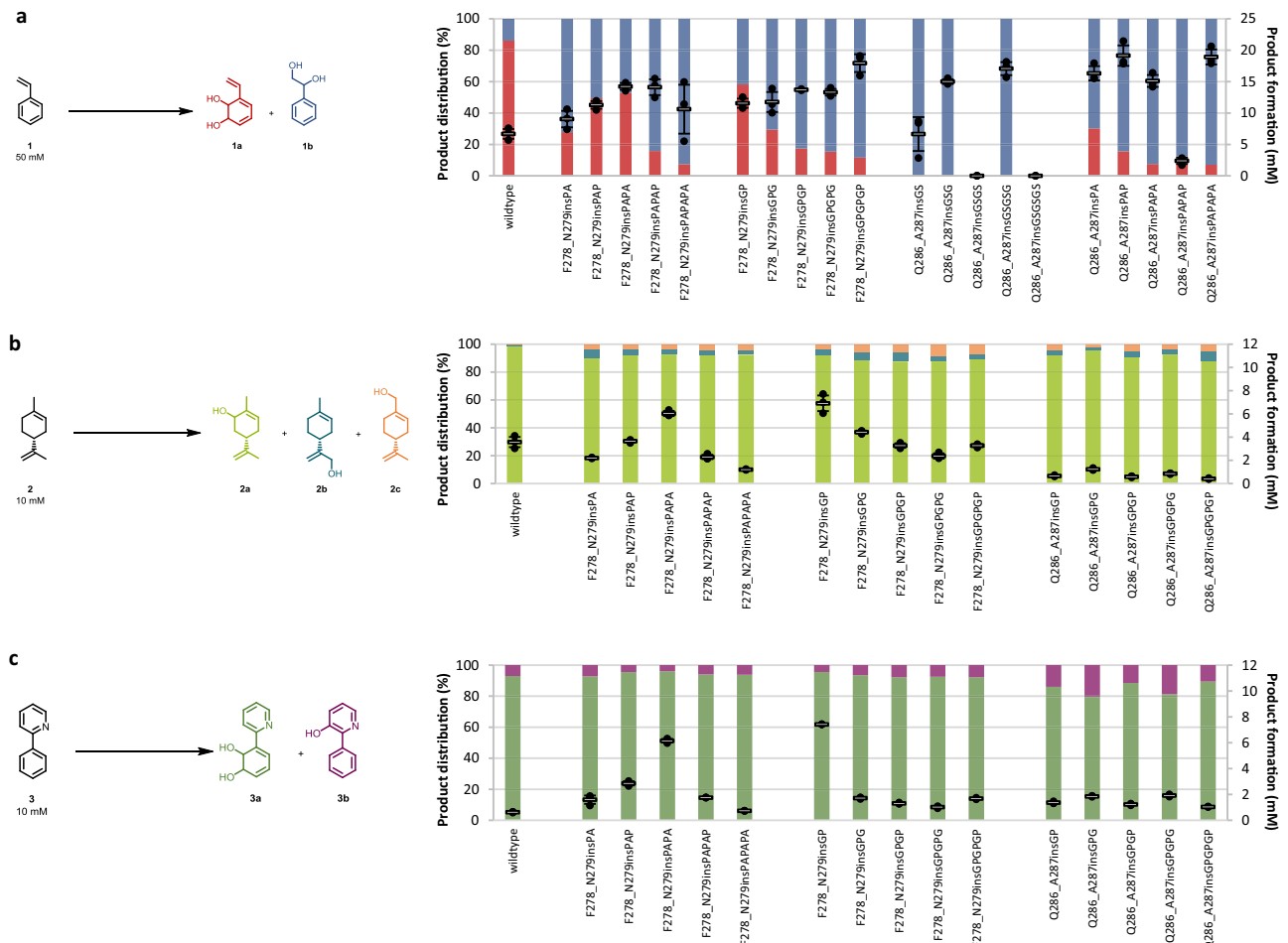

**Fig. 5 Biotransformation results with selected LILI variants. a** Product formation and distribution of the products **1a** (red, bottom bars) and **1b** (blue, top bars) obtained during the biotransformation of **1** with selected variants of the LILI library. **b** Product formation and distribution of the products **2a** (green, bottom bars), **2b** (blue, middle bars) and **2c** (orange, top bars) obtained during the biotransformation of **2** with selected variants of the LILI library. **c** Product formation and distribution of the products **3a** (green, bottom bars) and **3b** (purple, top bars) obtained during the biotransformation of **3**. The reaction conditions as mentioned in Fig. 2 were applied, the substrate concentrations are indicated in the reaction equation. The reactions were performed in technical triplicates (black dots) with average values (horizontal bar) and standard deviations (calculated using Excel version 2016) indicated. Error bars may be covered by markers. Source data are provided as a Source Data file.

whole loop, can affect its dynamics, resulting in numerous more active or selective variants. The CDO revealed high robustness towards loop modulations with only 18% inactive InDel variants (10% overall), especially with the new LILI library at previously identified hot-spot positions. The expression of all variants was confirmed via SDS-PAGE, suggesting the loss of activity due to solubility or folding reasons. Interestingly, the widely used flexible "G"-insertions and "GS"-insertions did not always yield the best variants. The stiff "PA" and polar "GP" insertions and especially the uncommon deletions in loop 2 resulted in more successful variants. The obtained biocatalysts possess enhanced features, comparable to active-site mutations, shown by comparison in Supplementary Table 9. Halder et al. increased the product formation 1.2-fold for substrate **1** with the active site variant NDO_H295A, while Gally et al. achieved an 1.4-fold increase in product formation with a simultaneous inversion of the regioselectivity from 99.7% **1a** with the wild-type to >99% **1b** with CDO_M232A[17,18]. A similar selectivity switch was reported by Vila et al. for the active site variant TDO_T365N[29]. Comparable changes could be achieved by the InDel variants CDO_Q286_A287insGSGSG and CDO_F282_L284del regarding a switched regioselectivity, both with an at least doubled product formation, or the 3.4-fold increase in product formation with

CDO_N279_D280insE. Also for **2**, compared to the wild-type doubled product formations could be achieved either by active site mutations, shown by Gally or Halder, or by point mutations and deletions in loop 2[17,18]. Furthermore, the amount of **2b** and **2c** could be strongly increased by the mutations L284G or M289A. Boyd et al. compared the wild-types of TDO, NDO and a BPDO regarding the biotransformation of **3**[63]. The different yields could be due to differences in the loops, since ROs often have similar active pockets and only slightly conserved loops. Different loops in the CDO also resulted in drastic changes in the product formation (16-fold increase with CDO_A283del) and regioselectivity (20% **3b** with CDO_Q286_A287insGPG). This comparison between mutations in the active pocket and in loops shows that similar results can be achieved with both approaches. Besides increased activities and selectivities, unexpected changes in the ee and de values were obtained, demonstrating the possibilities of loop engineering. These changes suggest additional degrees of freedom of the substrate inside the active pocket. The CAVER analysis of the active site tunnels suggested possible increases in volume, enabling new orientations of the substrate towards the iron which could result in changes in regioselectivities and enantioselectivities, broadening of the substrate scope and possible faster intake and release of substrate and product.

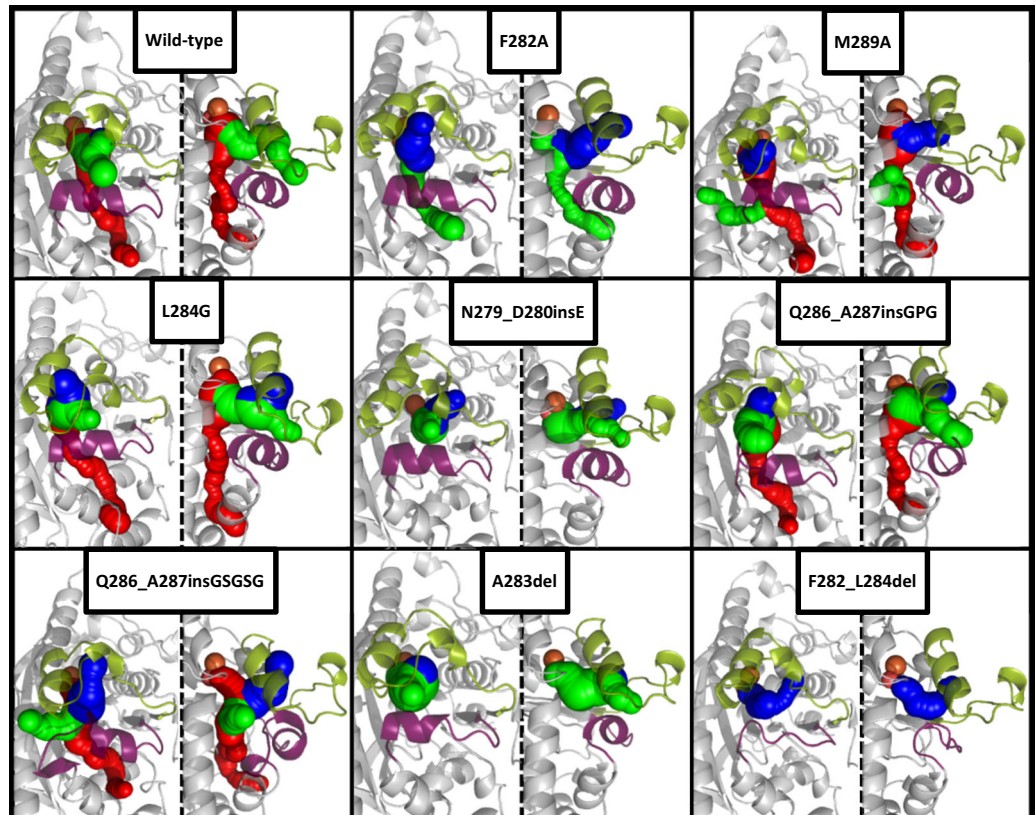

**Fig. 6 CAVER analysis of the crystal structure-based homology models of the wild-type and selected loop variants from two different perspectives.** The variants listed in Supplementary Table 9 were subject for homology modeling with subsequent MD refinement and 6 ps of MD simulations. The MD simulations were also performed on the crystal structure of the wild-type (PDB entry: 1WQL). The resulting energy minimized structures were analyzed with the CAVER plugin for PyMOL with a probe radius of 1.2 Å and iron as starting point. Loop 1 (green cartoon), loop 2 (purple), and the iron (orange sphere) are shown, as well as the resulting tunnels (tunnel 1 in blue, tunnel 2 in green, tunnel 3 in red).

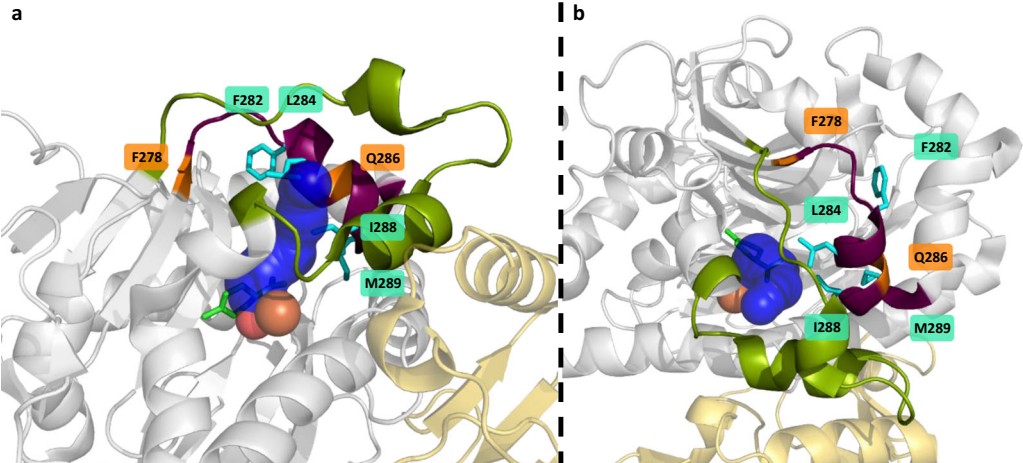

**Fig. 7 Highlighted positions with influence on product formation and selectivity by site mutation or insertion.** Side (**a**) and top view (**b**) of the α-subunit (gray) and β-subunit (yellow) of the CDO oxygenase (PDB entry: 1WQL). Loop 1 (green cartoon), loop 2 (purple), iron (orange sphere), oxygen (red spheres), and the docked *R*-(+)-limonene, as well as the via CAVER calculated active site tunnel (blue, probe radius 1.2 Å, iron as starting point) are shown. The residues F282, L284, and M289, whose site mutations had a major impact on selectivity and product formation, are highlighted as cyan sticks. The LILI-based insertion sites F278 and Q286 are highlighted as orange cartoons.

The comparison of the different approaches suggests a combination of alanine scan and LILI as a promising workflow for loop engineering. The selected positions were initial examples as insertions sites for LILI, additional positions especially in loop 2, in other loops of the CDO or in other ROs will be tested.

Furthermore, the combination with active-site variations will yield interesting results by first optimizing loops and then fine-tuning the orientation of the substrate in the pocket by active-site residue-engineering. Loop modulation can be a useful addition and extension of the toolbox of enzyme engineering.

## Methods

**Reagents, strains, plasmids, and genetic methods**. All utilized chemicals, except otherwise noted, were of analytical grade and obtained from Sigma-Aldrich (Steinheim, Germany), Carl Roth (Karlsruhe, Germany), Merck and VWR (Darmstadt, Germany) and Alfa Aesar (Karlsruhe, Germany). KOD Hot Start Polymerase Kit was obtained from Sigma-Aldrich (Steinheim, Germany), *Dpn*I from New England Biolabs GmbH (Frankfurt a.M., Germany), Kits for PCR product purification, gel extraction of DNA and minipreparations from ZYMO Research (Freiburg, Germany). Primers were synthesized by metabion International AG (Martinsried, Germany). Sequencing was performed by GATC Biotech (Konstanz, Germany) and Microsynth Seqlab (Göttingen, Germany). *Escherichia coli* XL1-Blue from Stratagene (Agilent, St. Clara, USA) was used for DNA manipulation, *E. coli* JM109(DE3)[17] from our in-house library, originated from Prof. Dr. Rebecca Parales, was used for expression and whole cell biotransformations. The plasmid pIP107D harboring the genes for CDO oxygenase, ferredoxin and reductase (Genbank: D37828.1) was described before (Supplementary Fig. 33)[17,64]. As empty vector, the closely related pUC19 was used (Supplementary Fig. 34)[64]. Point mutations, saturation mutagenesis, insertions and deletions were performed with the KOD-HotStart-Polymerase kit. The QuikChange mutagenesis PCR procedure was realized according to the user manual with the oligonucleotides given in Supplementary Tables 13–21. Mutations which could not be performed *via* QuikChange were achieved with overlap PCR and subsequent isothermal Gibson assembly[65]. Variants were confirmed by DNA sequencing.

**Enzyme expression**. Vectors harboring the wild-type and the different CDO variants were freshly transformed into *E. coli* JM109(DE3), plated onto selective agar plates, containing 100 µg/ml ampicillin and incubated overnight at 37 °C[66]. For the biotransformation of substrates **1**–**3** in 1 ml scale in 20 ml headspace vials, overnight cultures were grown in LB, which were then used to inoculate 100 ml TB Medium with 100 µg/ml ampicillin in 1 l flasks at 6% (*v/v*). The cultures were grown at 37 °C and 180 rpm in an orbital shaker until $OD_{600}$ reached 0.6–0.8. Expression was induced with 1 mM IPTG and then incubated at 25 °C at 180 rpm for 20 h. Harvesting was done by centrifugation ($4000 \times g$, 4 °C, 20 min) and the supernatant was discarded. For the saturation of individual residues via NNK-codon, the variants with saturated positions were transformed as described before. To achieve a library coverage of almost 95%, 92 wells of a 96 deep well plates, each containing 1 ml of TB medium with 100 µg/ml ampicillin, were inoculated with a single colony. Also, one well was inoculated with a colony harboring the empty vector, as well as three wells with the wild-type. The plate was covered with a Breathe Easier sealing membrane (Sigma-Aldrich, Steinheim, Germany) and incubated at 37 °C and 800 rpm for 20 h. 50 µl of each well were transferred to 950 µl TB medium with 100 µg/ml ampicillin in an additional plate which is incubated at 37 °C and 800 rpm for 4 h. To the first plate 500 µl 50% (*v/v*) glycerol are added and stored at −80 °C for mutation identification. The expression in the second plate in induced with 1 mM IPTG and incubated at 25 °C and 800 rpm for 20 h. The cells were harvested by centrifugation ($4000 \times g$, 20 min) and the supernatant was discarded.

**Biotransformation and extraction**. The in vivo biotransformation of the substrates **1** – **3** in 1 ml scale was carried out in gas-tight 20 ml headspace vials with an air–solution ratio of 19:1 to ensure enough oxygen intake[66]. The reaction mixture (1 ml) contained 0.1 $g_{cww}$/ml *E. coli* cells in 100 mM KPi buffer (pH 7.4), 20 mM glucose for cofactor regeneration and 50 mM styrene **1** or 10 mM (*R*)-(+)-limonene **2** or 2-phenylpyridine **3** respectively. Reactions were incubated at 30 °C and 180 rpm for 4 h. The reaction mixture was extracted with 1 ml MTBE with 1 mM biphenyl as internal standard, by 2 min vortexing and subsequent centrifugation ($8000 \times g$ rpm, RT, 5 min). The organic phase was transferred into GC vials for further analysis.

In vivo biotransformations of (*R*)-(+)-limonene in 96 deep well plates was carried out by the cell pellets from previously described expression, resuspended in 400 µl reaction mixture containing 100 mM KPi buffer (pH 7.4), 2% (*m/v*) (2-Hydroxypropyl)-β-cyclodextrin to reduce substrate evaporation, 20 mM glucose, 20 mM substrate and 100 µg/ml ampicillin. The plate was covered with a Breathe Easier sealing membrane and incubated at 25 °C and 300 rpm for 4 h. Resulting products and remaining substrate were extracted by the addition of 650 µl ethylacetate/cyclohexane mixture (1:1 (*v/v*)), containing 1 mM biphenyl as internal standard, to each well. The plate was vortexed for 3 min, centrifuged and the organic phase of each well transferred to an individual GC vial for further analysis.

**Analytical methods**. Optical density was measured at 600 nm in an Eppendorf BioPhotometer (Eppendorf, Hamburg, Germany).

The quantification of compound concentrations of the biotransformations of **1** and **2** in headspace vials was determined via GC FID using a Shimadzu GC-2010 GC system with FID detector, equipped with HP-1ms UI column (30 m × 0.25 mm × 0.25 µm, Agilent, Santa Clara, USA) and hydrogen as carrier gas at a linear velocity of 30 cm/s. The injector temperature was set to 250 °C and the compounds were detected with a flame ionization detector (FID) at 330 °C.

Injections of 1 µl were performed in split mode with a split of 1:15. For styrene **1**, the column was equilibrated at 80 °C for 3 min, then heated to 100 °C with a rate of 20 °C/min, then to 150 °C with 250 °C/min, then to 190 °C with 15 °C/min and finally with a rate of 250 °C/min to 320 °C, which was held for 3 min. For analysis of the biotransformation of (*R*)-(+)-limonene **2**, the column was equilibrated at 120 °C for 3 min, then heated to 190 °C with a rate of 13 °C/min and finally to 320 °C with 250 °C/min. the final temperature was held for 3 min.

Saturation mutagenesis of single residues with subsequent biotransformation of *R*-limonene **2** was analyzed via GC-MS using an Agilent 7820 A GC system with a 5977B mass spectrometer, equipped with a HP-5ms UI column (30 m x 0.25 mm × 0.25 µm, Agilent, Santa Clara, USA) and helium at a constant flow of 1.2 ml/min. The injector temperature was set to 250 °C with an injection volume of 1 µl in split mode with a ratio of 1:200. The compounds were detected by mass spectrometer with EI ionization with 70 eV, an ion source temperature of 230 °C and an interface temperature of 150 °C. Mass detection was performed in SIM mode with the selected ion masses of 91 m/z, 106 m/z, 109 m/z, 112 m/z, 136 m/z, 152 m/z and 170 m/z. The column was equilibrated for 1 min at 115 °C and then heated to 190 °C with a rate of 35 °C/min. This final temperature was held for 1 min.

The determination of the enantiomeric excess of the biotransformation of **2**, the already mentioned GC-FID system, equipped with a CP-Chirasil-Dex CB chiral column (30 m × 0.25 mm × 0.25 µm, Agilent, Santa Clara, USA), hydrogen as carrier gas with a linear velocity of 30 cm/s and a FID detector temperature of 250 °C, was used. The injector temperature was set to 230 °C with an injection volume of 1 µl in split mode with a ration of 1:10. The column was equilibrated for 2 min at 75 °C, then heated to 104 °C with a rate of 15 °C/min. The column was then heated to 110 °C with a rate of 0.2 °C/min and then with a rate of 50 °C/min to a final temperature of 180 °C, which was held for 3 min.

The analysis and quantification of the biotransformation of the substrate **3** were performed via LC/MS, using an Agilent 1260 Infinity LC system with a G4225A HiP degasser, a G1312B binary pump SL, G1367E high performance auto sampler, G1316A column compartment, G1315D diode array detector (DAD) and 6130 quadrupole mass spectrometer. For the biotransformation of **3**, the LC/MS was equipped with a reversed-phase Eclipse XDB C18 column (150 mm×4.6 mm, 5 µm particle size, Agilent, Santa Clara, USA). 1 µl sample was injected, separated with 1 ml/min of an isocratic mobile phase of 35% bi-destilled water and 65% methanol and a column temperature of 55 °C. The compounds were analyzed by DAD at a wavelength of 310 nm and via mass spectrometer with electron spray ionization (ESI) at a capillary voltage of 3500 V. Compound identification was performed in SCAN-SIM mode with positive polarity, a range of 70–300 *m/z* and the defined m/z ratios of 172 *m/z* and 190 *m/z*.

Determination of the enantiomeric excess of the biotransformations of **1** and **3** was executed via chiral HPLC on an Agilent 1200 HPLC system, equipped with a G1379B 1260 µ-Degasser, G1311A quaternary pump, G1329A auto sampler, G1316A column compartment and G1315D DAD. The separation was performed on a normal phase CHIRALPAK IB column (150 mm × 4.6 mm, 5 µm particle size, Daicel (Europa) GmbH, Raunheim, Germany). For the *ee* value of product **1b**, 15 µl sample were injected and the enantiomers were separated with 0.9 ml/min isocratic mobile phase of 98:2 n-hexane/ethanol at 20 °C. The analysis on the DAD was performed at 210 nm. The *ee* value of product **3a** was determined on the same HPLC system and column. 5 µl sample were injected and separated at 40 °C and an isocratic mobile phase of 80:20 n-hexane/ethanol with a flow of 0.7 ml/min. Analysis was performed using the DAD at a wavelength of 310 nm.

**In silico analysis**. The different variant primers were designed with SnapGene (Version 3.1.4). The substrate (*R*)-(+)-limonene **2** was docked into the crystal structure of the CDO wild-type (PDB entry: 1WQL) using the docking tools in YASARA (version 16.12.29)[67]. Sequence alignments were performed using the Clustal Omega online tool[68]. Phylogenetic trees were visualized with Iroki[69].

Changes in the loop structure of the variants shown in Fig. 6 were visualized by multiple in silico analysis steps. Based on the crystal structure of the wild-type (PDB entry: 1WQL), homology models of the selected variants were created using SWISS-MODEL[70]. The resulting homology models were subject for a 500 ps MD refinement for homology models in YASARA[59]. The resulting snapshot with the lowest energy was used for subsequent MD simulations of 5-6 ns length using the default settings, a water filled simulation cell with a density of 0.997 g/ml, 0.9% NaCl, pH 7.4 at 298 K[57]. The default Amber14 suite was used. The CAVER 3.0 plugin for PyMOL was applied on the average structure of all snapshots, calculated via YASARA[47]. Except for the radius which was set to 1.2 Å, the default settings were applied and the iron cation was set as starting point.

**Reporting summary**. Further information on research design is available in the Nature Research Reporting Summary linked to this article.

## Data availability

The data that support the findings of this study are available from the corresponding author upon request. Sequence (GenBank: D37828.1) and crystal structure (PDB entry: 1WQL) derived from Dong et al.[14]. Source data are provided with this paper.

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

## Acknowledgements

We cordially thank Dr. Bettina Nestl, Julian Wissner, Lea R. Rapp, and Dr. Max-Philipp Fischer for the fruitful discussions and support. We gratefully acknowledge the financial support by the Federal Ministry of Education and Research (BMBF) in the course of the project PowerCART (Funding number: 031B0369A).

## Author contributions

D.A. was involved in establishing the 96-DWP-Screening. P.M.H. performed the experimental studies and analytics. B.H. designed and supervised the work.

## Funding

## Competing interests

The authors declare no competing interests.
