## [Peer Review File · Nature Communications]

REVIEWER COMMENTS

Reviewer #1 (Remarks to the Author):

Loop remodeling has been proved to be a powerful means in enzyme engineering. The manuscript describes the loop engineering of a cumene dioxygenase from *Pseudomonas fluorescens* IP01 for the oxidation of three distinct substrates. A series of mutants with changes in chemo- and regioselectivity as well as activity were obtained. The manuscript may not suitable for the high quality required by Nat Commun, could be reconsidered after extensively revised and essential data added, see below.

Major comments:

1. The result sections are not well organized. Many mutagenesis methods used in this study, including alanine scan, site-specific saturation mutagenesis, insertion/deletion based on consensus, insertion/deletion based on liners, the mutants from these distinct strategies are not described clearly, jumping from one section to another, thereby confusing to catch up and lacking continuity. For instance, the section “Deletion of loop residues improves the CDO biocatalyst” should be better placed behind the section “Evaluation of an adaption library based on new oxygenases”, because both of them are talking about insertions and deletions based on consensus analysis, more logically connected. But in this current version, the two sections are separated by “Insertion of protein linkers as loop engineering strategy”, which is a distinct story based on protein linkers.
2. The stepwise loop insertion in LINDA is not a new approach, a highly similar catchy StLois, standing for stepwise loop insertion strategy, was previously published by Hoque et al. in 2017 (ACS Chem. Biol. 2017, 12, 1188-1193), this paper should be cited and discussed anyway.
3. The catalytic mechanisms of superior mutants have been analyzed very superficially. A single short MD trajectory in 5 ns does not really reveal clear changes in loop dynamics (check for instance: J. Chem. Theory Comput., 10.1021/acs.jctc.8b00391). Therefore, the

conclusions obtained from a single replica of 5 ns MD simulations could not be really accurate. More powerful computational analysis or crystallographic proof are needed.

Minor points to be addressed:

1. Lines 83-84, “they differ mainly by only one sequence region that encodes an active-site loop in other ROs”. Please clarify the difference is on the amino sequence level or on the 3D structural level, provide a sequence or structural alignment here can make a better representation. Moreover, what’s the relationship between the encoded “an active-site loop” and loop 1 and 2 described in the next paragraph?
2. Lines 91-92, “Loop 2 was identified as a hot-spot region with numerous improved variants.” Please add relative references to support it.
3. Line 93, it seems that the word “library” is not necessary in this sentence.
4. Line 120, change the word “usually” to “common”.
5. Line 124, delete the word “oxygenase”, which is duplicated with CDO. Add “the” before “two loops for this study”.
6. Line 127, add “values” after “The b-factor”.
7. Line 142, from where to the iron cation is 20 Å?
8. Line 147, the sentence here is nonsense, please rephrase.
9. Line 150, “...and M289 in particular” for what? please rephrase.
10. Line 151, change “compared to the wild-type” to “compared to that of the wild-type enzyme”.
11. Line 152, I guess what the authors did is single site specific saturation mutagenesis, not combinatorial or others, please specify it and also provide the info of the library size and coverage.
12. Line 157, the authors should provide the substrate concentration, otherwise readers will be confused with the number 98.2% shown in this sentence.
13. Line 158, change “regioselectivity to 97% 3a” to “regioselectivity upto 97% of 3a”.

14. Line 160, how did the authors conclude that “residue L284 influences the substrate binding and positioning” only from an Alanine scan experiment? Need more evidence to support the binding/positioning conclusion.

15. Line 161, please rephrase “The saturation variant L282G showed...” L282G is just a single site mutant, the word “saturation” used here is nonsense.

16. Line 163, rephrase the word “mutation-variants”.

17. Line 170, “This assumption of missing π - π stacking could hint the influence of these two positions on loop dynamics”, why influence on loop dynamics? It exists a huge gap between the observation and conclusion.

18. Considering Fig. 2 is placed in the first section of results and it is mainly corresponding to alanine scanning variants, the insertion variants part should be separated as an individual figure shown in the next section.

19. Line 193, please give the according reference(s) to this sentence.

20. Line 198, change the word “sequence” to “analysis”.

21. Line 211, “This result is consistent with the observations from the alanine scan”, why the result of F278G and D281E is consistent with that of alanine scan, since the latter only gives alanine mutants?

22. Line 215, change to “and at the end of loop 2”.

23. Line 217, cannot understand what the sentence “if changes... to altered enzyme properties” means.

24. Line 234, change “library” to “libraries”.

25. Line 247, change “while longer GP insertions lower the activity but reduce the ...” to “while longer GP insertions caused lower activity but reduced the ...”.

26. Line 253, “Consistent with the previous results, ...” what are the previous results, please specify.

27. Line 281, change “for” to “to construct”

28. Line 282, cannot understand the sentence “Then, one respectively three amino acids neighboring to the previously deleted residues were removed.” Please rephrase.

29. Line 286, again, “one respectively three amino acids” what does this mean?
30. Line 291, delete “compared to the wild-type switched”, or remove it to the end of this sentence.
31. Line 301, change “literature” to “literatures”.
32. Line 360, delete or rephrase “respectively glycine”.
33. Line 364, add “altering” after “thereby”.
34. Line 431, oC and h are units, should add a blank space between numbers and units.

Reviewer #2 (Remarks to the Author):

The paper describes a detailed exploration of the engineering of active site loops in RO enzymes. Starting from a phylogenetic analysis, and then alanine scanning of two loops, hotspots were identified with potential to diversify activity, specificity and enantioselectivity. An interesting approach was conceived, exploring saturation mutagenesis, then loop insertions and deletions, including the deliberate insertion of segments of altered flexibility. The new variants achieved a diverse range of significant improvements in activity, substrate selectivity and enantioselectivity, demonstrating the potency of the approach which could be generalised to other enzymes in future. Analysis of the reaction products was thorough. CAVER analysis was used on short MD simulations to gain some insight into how certain variants achieved their new properties. One key aspect was the modulation of tunnel access to the active site which would potentially alter the binding and access for substrates. Overall this is an exciting further development in the engineering of enzymes particularly through modification of active site loops.

Reviewer #3 (Remarks to the Author):

The paper submitted entitled “Active-Site loop variations adjust activity and selectivity of the cumene dioxygenase” by Heinemann et al. presents an interesting story about the Rieske non-heme iron oxygenase cumene dioxygenase. This enzyme is one of a large superfamily of enzymes that form synthetically valuable compounds and degrade aromatic pollutants. Intriguingly, this work pinpoints a region of cumene dioxygenase that impacts the activity and selectivity of the enzyme. Thus, this work has biocatalytic implications for engineering new compounds and uses some innovative engineering techniques to look at these so-called “hot spots”. Overall, the caliber of this work is appropriate for Nature Communications: the manuscript reveals thought-provoking findings about cumene dioxygenase and displays some interesting changes in activity. However, this manuscript needs some clarification, editing, and a few additional references. Once improvements are made, I do think this manuscript has really intriguing findings that will be of broad interest.

Suggested Improvements:

1. A few key references that similarly focus on the loop region of a Rieske oxygenase should be included (NdmA, dicamba monooxygenase, etc).
2. Figure 1 should include an overall figure of the Rieske structure so that one can orient themselves to where the loop region is actually located. It doesn't seem like the Rieske cluster is shown in this figure. The extra benefit of including an overall structure image is that it will allow the reader to think about the active site tunnels that have been calculated in other ROs and where it is in relation to those enzymes. An overall RO figure would likewise be helpful for thinking about the results in Fig, 4.
3. Figure 1c and Figure S1 are both hard to read as the letters are very blurry.
4. Line 74 talks about “comparison of ROs active-sites reveals high conservation of the same or similar, mainly hydrophobic amino acids”. This sentence seems pretty narrow in the context of a superfamily of enzymes that has more than 50,000 annotated members. It should highlight what ROs were looked at, and it might make sense to comment that less than 20 different ROs have been structurally characterized (many of which work on aromatic compounds).

5. Line 79 mentions 19 different Rieske oxygenases in *Phenylobacterium* and references the draft genome paper. It is worth commenting whether these oxygenases are annotated to have specific functions because it seems to be implied that their specificity comes from the loop region only. In addition to saying what is known about these proteins, the addition of a sequence alignment and/or a percent sequence identity would be helpful to motivate this sentence. I think these are actually shown in Fig. S1/S7 so they could be referenced.
6. Line 120 “compared to the usually planar structure of other ROs”, is this sentence referring to the substrates and if so, is it the dioxygenases that work on aromatic molecules only?
7. Line 127 mentions the B-factor of loop 2, which has one residue that goes up to 46 Å² “indicating high flexibility”. The inclusion of the average B-factor the whole structure would be beneficial for thinking about the flexibility of this loop.
8. With regards to point 6 about the loops and point 2 about the overall figure, it would be beneficial to mention whether the purple loop has a lower B-factor because it is packed up against another molecule in the crystal lattice or another subunit of the Rieske trimer.
9. An up-front introduction to cumene dioxygenase and the reactions it was known to catalyze and substrates it was known to accept prior to this work would facilitate a clear reading of this manuscript. It would also be helpful to highlight any interactions between these loops and the co-crystal structures of cumene dioxygenase – the presence or lack of interactions is also hard to see in Fig. 1.
10. A figure that highlights the identified variants that drastically affect the substrate positioning would be a helpful addition to the conclusion. The residues shown in Fig. 1 are hard to see.
11. For the variants that are reported to be inactive, a statement would be included to clarify whether the reason was loss of activity was substrate binding, solubility, folding, etc.
12. Line 315 states ‘tunnels play crucial roles in access and egress’, should also include references to RO papers that have calculated tunnels and addressed their importance such as the 2017 paper on naphthalene dioxygenase.

Reviewer #4 (Remarks to the Author):

Dear Colleagues,

I am writing to provide a review of Nat. Commun. manuscript NCOMMS-20-36068-T. Please provide my signed review in full to the authors. It is my belief that taking ownership of a review makes it more thoughtful, more objective, and more collegial.

In this study, the authors explore the use of different mutagenesis approaches to improve the activity and selectivity of a cumene dioxygenase (CDO). The authors first conduct an alanine scan of two loops of CDO and evaluate the activity of the resulting variants on three substrates to capture changes in CDO oxygenation activity and selectivity, including fascinating examples of pyridine hydroxylation. Significant changes in activity and selectivity were observed even for point mutations distal to the active site. Residues 278 and 280-289 in these loops were subjected to saturation mutagenesis and additional improved variants were obtained (e.g. mutations at F278, F282 and M289). The authors conclude this section by stating that “loop flexibility should be included for additional engineering approaches by InDels in order to obtain equivalent or even better biocatalysts compared to the alanine scan and saturation variants.” While reasonable, this passage should be rephrased to emphasize that it is a suggestion consistent with the reported findings rather than a strong conclusion based on specific evidence.

A separate mutagenesis approach based on sequence alignments of different Rieske oxygenases was then taken to investigate indels. One library involved point mutations or deletions, while the other involved different length insertions (note there is a typo in this sentence on p9, line 199). Again, different activities and selectivities are observed, but it is specifically noted that a majority of the insertion mutants are active, indicating fairly high tolerance of insertions by CDO. Up to this point, all of the mutagenesis strategies reported are routine, but what I took to be the key point of the paper is introduced at the bottom of page 9. Specifically, the authors posit that “stepwise deletions or insertions of amino acids with known dynamic properties can yield a deeper understanding how altered loop agilities could affect enzyme properties”. While many indel mutagenesis strategies exist, I can’t think of an obvious example that tries to look at wellcharacterized linkers in the way that the authors describe. As emphasized below, it seems to me that emphasizing this point of novelty earlier would more effectively communicate this aspect of the manuscript, though that also depends on how the approach worked out, which is currently difficult to evaluate. The authors

implement this approach at three sites (V257, F278, and Q286) identified based on results of either point mutation or initial insertion libraries (based on a different insertion approach). Again, different activities and selectivities are observed, a compilation of mutation effects are presented in Table 1, and simulations are used to suggest possible mechanisms by which mutations alter catalysis.

At the outset of the manuscript, the authors note the potential importance of point mutations, insertions, and deletions in active site loop regions on enzyme catalysis. References for all of these possibilities are provided, and the current study again shows this to be the case. As suggested above, I believe that conflating these different approaches detracts from the conceptual novelty of the current study (LINDA mutagenesis) and ends up making it difficult (at least for me) to follow its key lessons. Focusing the manuscript entirely on indel mutagenesis (perhaps using alanine scanning as a means of identifying sites to target) would help in this regard.

The key findings from the LINDA process are summarized in Figure 3 and p10. In the text, the authors state that “in comparison to the alignment-based library, LINDA variants revealed interesting trends based on the length and composition of the insertions, mainly after F278 and Q286,” but the trends highlighted on p10 seem just as perplexing as others reported for point mutations and other indel libraries. For example, the authors state that “longer insertions tend to increase the formation of 1b,” but group 2 insertions at Q286_A287 gave exclusively 1b even for the shortest linkers examined (Fig. 3b). The authors note that “a single inserted residue can make the difference between increased activity and none,” but this set of variants is only shown for the reaction of 1, so it is unclear if this is a general trend. While large changes in selectivity were observed for substrate 1, only minimal changes in selectivity and moderate (up to 2x) changes in yield were observed for substrate 2. Significant increases in yields were observed for substrate 3, and while the authors highlight the importance of Q286_A287 insertions on selectivity (as for substrate 1), greater effects on selectivity for substrate 3 are observed for F278_N279 insertions (Fig. 3d). I also don’t understand why the authors note “Q286_A287insGPG with up to 20.2% 3b and 18.6% product formation” when the same selectivity and nearly 80% yield are obtained with F278_N279insGP. I may be misreading the text or tables, but if not, this section needs significant revision to account for the discrepancies noted above.

Because of the issues outlined above, it is difficult for me to verify that the authors achieved their stated goal of revealing “deeper understanding how altered loop agilities could affect

enzyme properties”. If the authors wish to maintain an emphasis on indel mutagenesis, I think a clearer presentation of any trends will be necessary. In general, it does not appear that Group 1 linkers were used for any of the results in Figure 3, so these should be moved to the SI. It would also be helpful if the data were presented in a way that apples-to-apples comparison between the effects of each linker on each substrate is possible, ideally with b/c/d in a vertical stack with identical libraries aligned (like Figure 2, which is quite easy to follow). It appears that there is a consistent increase and then decrease in conversion for all three products (and even a loose selectivity trend for substrates 1 and 3) based on linker length for the F278_N279 Group 3 linker library, and trends like this would be easier to see in such a format.

The results of all mutations are summarized in Table 1, which shows that “different engineering approaches yielded various loop variants with changed biocatalytically important properties”. This conclusion would be true, however, for any mutagenesis approach, so a more specific conclusion would be helpful to direct the reader’s focus. The table is also quite difficult to navigate, as it includes information from different mutagenesis approaches from this study and the literature, mutations outside of the loop regions that were the focus of the study, different culture conditions, and different substrates. If the authors wish to report both point mutations and indels, it would be helpful if a more concise comparison of the effects of point mutations and indels was presented. Currently, it is not clear that the insertion approach was any more effective than point mutagenesis (I appreciate this was not a stated goal, but readers will be curious), and it is difficult to compare the magnitude of improvements for each substrate over what was previously reported in the literature. This would be particularly useful where significant changes in regio/site selectivity were observed, though the best enzymes in this regard (1b, CDO_M232A, active site; 2b, CDO_L284G, loop; 3b, CDO_Q286_A287insGPG, loop) highlight the importance of different mutagenesis sites and strategies.

Finally, the authors use homology modeling and short MD simulations to gain insight into the structural effects of different mutations. In my opinion, great caution is warranted when drawing conclusions about enzyme catalysis from homology models, particularly involving flexible loop regions. I believe the section detailing these results should be reworded to indicate that the observed structural changes are “suggestive” or some other language along those lines. This conclusion, for example, is too strong based on the results presented; “The results of the CAVER analysis underline how loop modulations alter active site tunnels and

provide first hints how the incorporated mutations influence catalytically important properties of the enzymes despite their big distance to the iron.”

In summary, while I think the authors make a strong case that indel libraries remain a useful mutagenesis strategy, the presentation is difficult to follow and doesn't seem to deliver on the goal of providing an improved means to rationalize the effects of indels on enzyme catalysis. A significant revision focused on either indel mutagenesis or the more general goal of changing CDO selectivity/scope with additional depth in each case to establish either the unique potential of the LINDA mutagenesis strategy or high levels of selectivity for products not previously reported might be able to address these issues.

Sincerely,

Reply to Reviewer's comments

Reviewer #1 (Remarks to the Author):

Loop remodeling has been proved to be a powerful means in enzyme engineering. The manuscript describes the loop engineering of a cumene dioxygenase from *Pseudomonas fluorescens* IP01 for the oxidation of three distinct substrates. A series of mutants with changes in chemo- and regioselectivity as well as activity were obtained. The manuscript may not suitable for the high quality required by Nat Commun, could be reconsidered after extensively revised and essential data added, see below.

Many thanks for the detailed suggestions and recommendations for improvement.

Major comments:

1. The result sections are not well organized. Many mutagenesis methods used in this study, including alanine scan, site-specific saturation mutagenesis, insertion/deletion based on consensus, insertion/deletion based on liners, the mutants from these distinct strategies are not described clearly, jumping from one section to another, thereby confusing to catch up and lacking continuity. For instance, the section "Deletion of loop residues improves the CDO biocatalyst" should be better placed behind the section "Evaluation of an adaption library based on new oxygenases", because both of them are talking about insertions and deletions based on consensus analysis, more logically connected. But in this current version, the two sections are separated by "Insertion of protein linkers as loop engineering strategy", which is a distinct story based on protein linkers.

Thank you for the constructive suggestions. In accordance to this, we changed the order and rephrased multiple sections to ease the follow-up of the study.

2. The stepwise loop insertion in LINDA is not a new approach, a highly similar catchy StLois, standing for stepwise loop insertion strategy, was previously published by Hoque et al. in 2017 (ACS Chem. Biol. 2017, 12, 1188-1193), this paper should be cited and discussed anyway.

We added a short discussion about the similarity between the very interesting StLois approach by Hoque *et al.* and our adaption library based on the sequence alignment. Furthermore, a

short remark addressed the differences between the two approaches and the LINDA (now LILI) approach.

3. The catalytic mechanisms of superior mutants have been analyzed very superficially. A single short MD trajectory in 5 ns does not really reveal clear changes in loop dynamics (check for instance: J. Chem. Theory Comput., 10.1021/acs.jctc.8b00391). Therefore, the conclusions obtained from a single replica of 5 ns MD simulations could not be really accurate. More powerful computational analysis or crystallographic proof are needed.

Thank you for this very interesting reference and the critique about the too strong statement of our short, more suggestive simulations. In accordance to your comment and reviewer 4, we changed the wording of the simulation section and added a statement about the inaccuracy of our simulations.

Minor points to be addressed:

1. Lines 83-84, “they differ mainly by only one sequence region that encodes an active-site loop in other ROs”. Please clarify the difference is on the amino sequence level or on the 3D structural level, provide a sequence or structural alignment here can make a better representation. Moreover, what’s the relationship between the encoded “an active-site loop” and loop 1 and 2 described in the next paragraph?

We linked the description of the oxygenase-sequences from *Phenylobacterium immobile* E to the sequence alignment in Figure S1 and clarified the relation between this sequence space and the loops in the CDO. Additionally, we added a reference to naphthalene dioxygenase, which active site loop is also coded in this sequence space.

2. Lines 91-92, “Loop 2 was identified as a hot-spot region with numerous improved variants.” Please add relative references to support it.

We changed the sentence accordingly to emphasize that loop 2 was identified as hot-spot region by the different applied engineering approaches in the course of this study.

3. Line 93, it seems that the word “library” is not necessary in this sentence.

It was changed accordingly.

4. Line 120, change the word “usually” to “common”.

It was changed accordingly.

5. Line 124, delete the word “oxygenase”, which is duplicated with CDO. Add “the” before “two loops for this study”.

We performed the sequence alignment only with the α -subunits of the oxygenase of the CDO, which is a three-component system. We added “oxygenases” after ‘ROs’ to emphasize our focus on the α -subunits of the oxygenases. We also added “the” accordingly.

6. Line 127, add “values” after “The b-factor”.

It was changed accordingly.

7. Line 142, from where to the iron cation is 20 Å?

We added D280, as the residues with the biggest distance (20.2 Å) to the iron cation, to the sentence.

8. Line 147, the sentence here is nonsense, please rephrase.

We rephrased the sentence to emphasize the influence of the mutations F282A and M289A.

9. Line 150, “...and M289 in particular” for what? please rephrase.

The sentence was extended by highlighting the increased product formation, as shown before with the variants F282A and M289A for substrate 2.

10. Line 151, change “compared to the wild-type” to “compared to that of the wild-type enzyme”.

It was changed accordingly.

11. Line 152, I guess what the authors did is single site specific saturation mutagenesis, not combinatorial or others, please specify it and also provide the info of the library size and coverage.

The applied degenerated codon, library size and coverage was added.

12. Line 157, the authors should provide the substrate concentration, otherwise readers will be confused with the number 98.2% shown in this sentence.

The used substrate concentrations for the biotransformations in 1 ml scale are provided in the Methods section. We furthermore added the used concentrations accordingly.

13. Line 158, change “regioselectivity to 97% 3a” to “regioselectivity upto 97% of 3a”.

It was changed accordingly.

14. Line 160, how did the authors conclude that ‘residue L284 influences the substrate binding and positioning’ only from an Alanine scan experiment? Need more evidence to support the binding/positioning conclusion.

We deleted ‘binding’ and added a more detailed explanation why we came up with the assumption of the influence of L284 on the substrate positioning.

15. Line 161, please rephrase ‘The saturation variant L282G showed...’ L282G is just a single site mutant, the word ‘saturation’ used here is nonsense.

In the course of the correction of comment 14, we corrected this accordingly.

16. Line 163, rephrase the word ‘mutation-variants’.

It was changed accordingly.

17. Line 170, ‘This assumption of missing π - π stacking could hint the influence of these two positions on loop dynamics’, why influence on loop dynamics? It exists a huge gap between the observation and conclusion.

This section was deleted due to the mentioned gap between the observation and the conclusion. Furthermore, in order to make it easier for the reader to follow, this slightly misleading results were removed.

18. Considering Fig. 2 is placed in the first section of results and it is mainly corresponding to alanine scanning variants, the insertion variants part should be separated as an individual figure shown in the next section.

We removed the results of the biotransformations with the insertion variants and combined them with deletion variants in a new figure, placed after the section about deletion variants.

19. Line 193, please give the according reference(s) to this sentence.

Two according references were added.

20. Line 198, change the word ‘sequence’ to ‘analysis’.

The incorporated point mutations were based on the consensus sequence, which resulted from the sequence analysis. Therefore, in our opinion, ‘consensus sequence’ is the better expression

21. Line 211, “This result is consistent with the observations from the alanine scan”, why the result of F278G and D281E is consistent with that of alanine scan, since the latter only gives alanine mutants?

We added the explanation that the mutation of these positions showed influences on the biocatalysis, as shown before in the alanine scan.

22. Line 215, change to “and at the end of loop 2”.

It was changed accordingly.

23. Line 217, cannot understand what the sentence “if changes... to altered enzyme properties” means.

This difficult to understand written sentence was amended accordingly.

24. Line 234, change “library” to “libraries”.

It was changed accordingly.

25. Line 247, change “while longer GP insertions lower the activity but reduce the ...” to “while longer GP insertions caused lower activity but reduced the ...”.

It was changed accordingly.

26. Line 253, “Consistent with the previous results, ...” what are the previous results, please specify.

We added the link to the sequence alignment-based insertions

27. Line 281, change “for” to “to construct”

It was changed accordingly.

28. Line 282, cannot understand the sentence “Then, one respectively three amino acids neighboring to the previously deleted residues were removed.” Please rephrase.

This misleading sentence was replaced by a more detailed description of the library construction.

29. Line 286, again, “one respectively three amino acids” what does this mean?

“respectively” was replaced by “or”.

30. Line 291, delete “compared to the wild-type switched”, or remove it to the end of this sentence.

We moved this part slightly modified to the end of the sentence.

31. Line 301, change “literature” to “literatures”.

It was changed accordingly.

32. Line 360, delete or rephrase “respectively glycine”.

We rephrased this sentence.

33. Line 364, add “altering” after “thereby”.

We added “altering”.

34. Line 431, oC and h are units, should add a blank space between numbers and units.

We checked the whole method section for missing blank spaces between numbers and units.

Reviewer #2 (Remarks to the Author):

The paper describes a detailed exploration of the engineering of active site loops in RO enzymes. Starting from a phylogenetic analysis, and then alanine scanning of two loops, hotspots were identified with potential to diversify activity, specificity and enantioselectivity. An interesting approach was conceived, exploring saturation mutagenesis, then loop insertions and deletions, including the deliberate insertion of segments of altered flexibility. The new variants achieved a diverse range of significant improvements in activity, substrate selectivity and enantioselectivity, demonstrating the potency of the approach which could be generalised to other enzymes in future. Analysis of the reaction products was thorough. CAVER analysis was used on short MD simulations to gain some insight into how certain variants achieved their new properties. One key aspect was the modulation of tunnel access to the active site which would potentially alter the binding and access for substrates. Overall this is an exciting further development in the engineering of enzymes particularly through modification of active site loops.

Reviewer #3 (Remarks to the Author):

The paper submitted entitled “Active-Site loop variations adjust activity and selectivity of the cumene dioxygenase” by Heinemann et al. presents an interesting story about the Rieske non-heme iron oxygenase cumene dioxygenase. This enzyme is one of a large superfamily of enzymes that form synthetically valuable compounds and degrade aromatic pollutants. Intriguingly, this work pinpoints a region of cumene dioxygenase that impacts the activity and selectivity of the enzyme. Thus, this work has biocatalytic implications for engineering new compounds and uses some innovative engineering techniques to look at these so-called “hot spots”. Overall, the caliber of this work is appropriate for Nature Communications: the manuscript reveals thought-provoking findings about cumene dioxygenase and displays some interesting changes in activity. However, this manuscript needs some clarification, editing, and a few additional references. Once improvements are made, I do think this manuscript has really intriguing findings that will be of broad interest.

Suggested Improvements:

1. A few key references that similarly focus on the loop region of a Rieske oxygenase should be included (NdmA, dicamba monooxygenase, etc).

Thank you for the fruitful suggestions. We added some more references to underline the importance of loop structures in Rieske oxygenases.

2. Figure 1 should include an overall figure of the Rieske structure so that one can orient themselves to where the loop region is actually located. It doesn't seem like the Rieske cluster is shown in this figure. The extra benefit of including an overall structure image is that it will allow the reader to think about the active site tunnels that have been calculated in other ROs and where it is in relation to those enzymes. An overall RO figure would likewise be helpful for thinking about the results in Fig, 4.

We added pictures of the hexameric and the dimeric quaternary structure of the oxygenase to Figure 1.

3. Figure 1c and Figure S1 are both hard to read as the letters are very blurry.

We replaced the pictures with alignments in higher resolution.

4. Line 74 talks about “comparison of ROs active-sites reveals high conservation of the same or similar, mainly hydrophobic amino acids”. This sentence seems pretty narrow in the context of a superfamily of enzymes that has more than 50,000 annotated members. It should highlight what ROs were looked at, and it might make sense to comment that less than 20 different ROs have been structurally characterized (many of which work on aromatic compounds).

We modified this phrase and linked it to comparison of active site residues of eight structurally characterized ROs by Ferraro *et al.* (Ferraro, D. J., Gakhar, L., & Ramaswamy, S. (2005). Rieske business: Structure-function of Rieske non-heme oxygenases. *Biochemical and Biophysical Research Communications*, 338(1), 175–190.)

5. Line 79 mentions 19 different Rieske oxygenases in *Phenylobacterium* and references the draft genome paper. It is worth commenting whether these oxygenases are annotated to have specific functions because it seems to be implied that their specificity comes from the loop region only. In addition to saying what is known about these proteins, the addition of a sequence alignment and/or a percent sequence identity would be helpful to motivate this sentence. I think these are actually shown in Fig. S1/S7 so they could be referenced.

We referenced the results from a proteome analysis performed in previous studies at our institute.

6. Line 120 “compared to the usually planar structure of other ROs”, is this sentence referring to the substrates and if so, is it the dioxygenases that work on aromatic molecules only?

We checked this line and in the manuscript, the referred phrase is “Compared to the usually planar substrates of other ROs, cumene as the natural substrate of the 118 CDO from *Pseudomonas fluorescens* IP01 is sterically more demanding,..” The mentioned planar structure refers to the substrate. We added a phrase about the broad substrate scope with the corresponding literature from Gally *et al.* (Gally, C., Nestl, B. M., & Hauer, B. (2015). Engineering Rieske Non-Heme Iron Oxygenases for the Asymmetric Dihydroxylation of Alkenes. *Angewandte Chemie - International Edition*, 54(44), 12952–12956.)

7. Line 127 mentions the B-factor of loop 2, which has one residue that goes up to 46 Å² “indicating high flexibility”. The inclusion of the average B-factor the whole structure would be beneficial for thinking about the flexibility of this loop.

More information about the b-factor values of the two loops and the whole α -subunit were added.

8. With regards to point 6 about the loops and point 2 about the overall figure, it would be beneficial to mention whether the purple loop has a lower B-factor because it is packed up against another molecule in the crystal lattice or another subunit of the Rieske trimer.

We added a sentence about the localization of the loops at the outer surface and referred it to the modified Figure 1.

9. An up-front introduction to cumene dioxygenase and the reactions it was known to catalyze and substrates it was known to accept prior to this work would facilitate a clear reading of this manuscript. It would also be helpful to highlight any interactions between these loops and the co-crystal structures of cumene dioxygenase – the presence or lack of interactions is also hard to see in Fig. 1.

A short introduction of previously catalyzed reactions and accepted substrates was added.

10. A figure that highlights the identified variants that drastically affect the substrate positioning would be a helpful addition to the conclusion. The residues shown in Fig. 1 are hard to see.

We added Fig. 6 with the most interesting positions for site mutagenesis and insertion highlighted.

11. For the variants that are reported to be inactive, a statement would be included to clarify whether the reason was loss of activity was substrate binding, solubility, folding, etc.

We added a short statement about this subject to the SI after Figure S11 which shows the SDS-PAGE of selected variants. We also added a sentence to the discussion.

12. Line 315 states ‘tunnels play crucial roles in access and egress’, should also include references to RO papers that have calculated tunnels and addressed their importance such as the 2017 paper on naphthalene dioxygenase.

We added the reference about the naphthalene dioxygenase as well as a reference about two dioxygenases involved in biosynthesis of paralytic shellfish toxins, where Lukowski *et al* highlight their active site tunnels.

Reviewer #4 (Remarks to the Author):

Dear Colleagues,

I am writing to provide a review of Nat. Commun. manuscript NCOMMS-20-36068-T. Please provide my signed review in full to the authors. It is my belief that taking ownership of a review makes it more thoughtful, more objective, and more collegial.

Thank you for this very detailed and fruitful review. We are very grateful about the effort and the openness of this review!

In this study, the authors explore the use of different mutagenesis approaches to improve the activity and selectivity of a cumene dioxygenase (CDO). The authors first conduct an alanine scan of two loops of CDO and evaluate the activity of the resulting variants on three substrates to capture changes in CDO oxygenation activity and selectivity, including fascinating examples of pyridine hydroxylation. Significant changes in activity and selectivity were observed even for point mutations distal to the active site. Residues 278 and 280-289 in these loops were subjected to saturation mutagenesis and additional improved variants were obtained (e.g. mutations at F278, F282 and M289). The authors conclude this section by stating that ‘loop flexibility should be included for additional engineering approaches by InDels in order to obtain equivalent or even better biocatalysts compared to the alanine scan and saturation variants.’ While reasonable, this passage should be rephrased to emphasize that it is a suggestion consistent with the reported findings rather than a strong conclusion based on specific evidence.

We rephrased and extended this sentence to a more suggestion-like phrase, that flexibility modulations could be a possible addition to enzyme engineering.

A separate mutagenesis approach based on sequence alignments of different Rieske oxygenases was then taken to investigate indels. One library involved point mutations or deletions, while the other involved different length insertions (note there is a typo in this

sentence on p9, line 199). Again, different activities and selectivities are observed, but it is specifically noted that a majority of the insertion mutants are active, indicating fairly high tolerance of insertions by CDO. Up to this point, all of the mutagenesis strategies reported are routine, but what I took to be the key point of the paper is introduced at the bottom of page 9. Specifically, the authors posit that “stepwise deletions or insertions of amino acids with known dynamic properties can yield a deeper understanding how altered loop agilities could affect enzyme properties”. While many indel mutagenesis strategies exist, I can’t think of an obvious example that tries to look at wellcharacterized linkers in the way that the authors describe. As emphasized below, it seems to me that emphasizing this point of novelty earlier would more effectively communicate this aspect of the manuscript, though that also depends on how the approach worked out, which is currently difficult to evaluate. The authors implement this approach at three sites (V257, F278, and Q286) identified based on results of either point mutation or initial insertion libraries (based on a different insertion approach). Again, different activities and selectivities are observed, a compilation of mutation effects are presented in Table 1, and simulations are used to suggest possible mechanisms by which mutations alter catalysis.

In agreement with the suggestion of Reviewer 1, we moved the subchapter “Deletion of loop residues improves the CDO biocatalyst” behind the chapter about the sequence alignment-based library. This should emphasize and highlight the linker insertions as new approach. Furthermore, we separated the linker insertions from deletions and renamed the approach to Linker In Loop Insertion (LILI).

At the outset of the manuscript, the authors note the potential importance of point mutations, insertions, and deletions in active site loop regions on enzyme catalysis. References for all of these possibilities are provided, and the current study again shows this to be the case. As suggested above, I believe that conflating these different approaches detracts from the conceptual novelty of the current study (LINDA mutagenesis) and ends up making it difficult (at least for me) to follow its key lessons. Focusing the manuscript entirely on indel mutagenesis (perhaps using alanine scanning as a means of identifying sites to target) would help in this regard.

We rephrased multiple sections to ease the follow up. The main text is now more or less divided in the sections alanine screen and saturation, sequence alignment-based InDels, deletions in flexible loops, the LILI approach as novel approach based on the previous sections, and further insights and possible changes in the tunnels analyzed *via* CAVER.

The key findings from the LINDA process are summarized in Figure 3 and p10. In the text, the authors state that “in comparison to the alignment-based library, LINDA variants revealed interesting trends based on the length and composition of the insertions, mainly after F278 and Q286,” but the trends highlighted on p10 seem just as perplexing as others reported for point mutations and other indel libraries. For example, the authors state that “longer insertions tend to increase the formation of 1b,” but group 2 insertions at Q286_A287 gave exclusively 1b even for the shortest linkers examined (Fig. 3b). The authors note that “a single inserted residue can make the difference between increased activity and none,” but this set of variants is only shown for the reaction of 1, so it is unclear if this is a general trend. While large changes in selectivity were observed for substrate 1, only minimal changes in selectivity and moderate (up to 2x) changes in yield were observed for substrate 2. Significant increases in yields were observed for substrate 3, and while the authors highlight the importance of Q286_A287 insertions on selectivity (as for substrate 1), greater effects on selectivity for substrate 3 are observed for F278_N279 insertions (Fig. 3d). I also don't understand why the authors note “Q286_A287insGPG with up to 20.2% 3b and 18.6% product formation” when the same selectivity and nearly 80% yield are obtained with F278_N279insGP. I may be misreading the text or tables, but if not, this section needs significant revision to account for the discrepancies noted above.

We addressed the discrepancies and rephrased various sentences in this section, regarding a more precise and detailed description of the results.

Because of the issues outlined above, it is difficult for me to verify that the authors achieved their stated goal of revealing “deeper understanding how altered loop agilities could affect enzyme properties”. If the authors wish to maintain an emphasis on indel mutagenesis, I think a clearer presentation of any trends will be necessary. In general, it does not appear that Group 1 linkers were used for any of the results in Figure 3, so these should be moved to the SI. It would also be helpful if the data were presented in a way that apples-to-apples comparison between the effects of each linker on each substrate is possible, ideally with b/c/d in a vertical stack with identical libraries aligned (like Figure 2, which is quite easy to follow). It appears that there is a consistent increase and then decrease in conversion for all three products (and even a loose selectivity trend for substrates 1 and 3) based on linker length for the F278_N279 Group 3 linker library, and trends like this would be easier to see in such a format.

Unfortunately, the proposed vertically stacked alignment of the results of the libraries would be hardly visible as normal figure in the main text, even though we completely agree with this suggestion. We added figure S11 in the SI, three stacked column charts like in Figure 3, in landscape orientation to ensure good visibility. We also referred to this figure in the main text about visible trends. We stick to Figure 4 (previously Fig. 3) as the highlights of the LLI approach.

The results of all mutations are summarized in Table 1, which shows that “different engineering approaches yielded various loop variants with changed biocatalytically important properties”. This conclusion would be true, however, for any mutagenesis approach, so a more specific conclusion would be helpful to direct the reader’s focus. The table is also quite difficult to navigate, as it includes information from different mutagenesis approaches from this study and the literature, mutations outside of the loop regions that were the focus of the study, different culture conditions, and different substrates. If the authors wish to report both point mutations and indels, it would be helpful if a more concise comparison of the effects of point mutations and indels was presented. Currently, it is not clear that the insertion approach was any more effective than point mutagenesis (I appreciate this was not a stated goal, but readers will be curious), and it is difficult to compare the magnitude of improvements for each substrate over what was previously reported in the literature. This would be particularly useful where significant changes in regio/site selectivity were observed, though the best enzymes in this regard (1b, CDO_M232A, active site; 2b, CDO_L284G, loop; 3b, CDO_Q286_A287insGPG, loop) highlight the importance of different mutagenesis sites and strategies.

Table 1 was moved to the SI and a more detailed discussion of the variants from this study compared to variants known from literature was added.

Finally, the authors use homology modeling and short MD simulations to gain insight into the structural effects of different mutations. In my opinion, great caution is warranted when drawing conclusions about enzyme catalysis from homology models, particularly involving flexible loop regions. I believe the section detailing these results should be reworded to indicate that the observed structural changes are “suggestive” or some other language along those lines. This conclusion, for example, is too strong based on the results presented; “The results of the CAVER analysis underline how loop modulations alter active site tunnels and provide first hints how the incorporated mutations influence catalytically important properties of the enzymes despite their big distance to the iron.”

We fully agree with the comment of the reviewer. The statements we made were too strong regarding the simulations, which were only meant to have a suggestion character. We have changed the wording and added a statement about the validity of our simulations.

In summary, while I think the authors make a strong case that indel libraries remain a useful mutagenesis strategy, the presentation is difficult to follow and doesn't seem to deliver on the goal of providing an improved means to rationalize the effects of indels on enzyme catalysis. A significant revision focused on either indel mutagenesis or the more general goal of changing CDO selectivity/scope with additional depth in each case to establish either the unique potential of the LINDA mutagenesis strategy or high levels of selectivity for products not previously reported might be able to address these issues.

Sincerely,

Reviewer #1 (Remarks to the Author):

I am happy with the extensively revised manuscript, could be accepted for Nature Commun.

Reviewer #3 (Remarks to the Author):

The changes to the order of this manuscript and inclusion of extra details and references have improved this manuscript. Overall, the manuscript would benefit from some careful editing, but offers interesting insights into the importance of Rieske oxygenase "hot-spots" for enzyme engineering efforts.

Reviewer #4 (Remarks to the Author):

The revised manuscript adequately addressed each of the issues that I brought up. Several instances in which conclusions were over-stated have been brought in line with experimental observations and computational limitations. The restructuring of the paper made it much simpler to follow. Most importantly, emphasis on how the insertion approach used can provide "deeper understanding" into the effects of insertions on activity has been somewhat lessened in favor of simply noting comparisons between insertion methods and mutagenesis methods. The expanded discussion makes this change particularly clear. I believe the manuscript is suitable for publication in Nature Communications.